# The Hot Mess of AI: How Does Misalignment Scale With Model Intelligence and Task Complexity?

**Alexander Hägele**[*1,2]     **Aryo Pradipta Gema**[1,3]     **Henry Sleight**[4]     **Ethan Perez**[5]

**Jascha Sohl-Dickstein**[*5]

[1]Anthropic Fellows Program [2]EPFL [3]University of Edinburgh [4]Constellation [5]Anthropic
[*]`alexander.hagele@epfl.ch`, `jascha@anthropic.com`

## Abstract

As AI becomes more capable, we entrust it with more general and consequential tasks. The risks from failure grow more severe with increasing task scope. It is therefore important to understand how extremely capable AI models will fail: Will they fail by systematically pursuing goals we do not intend? Or will they fail by being a hot mess, and taking nonsensical actions that do not further any goal? We operationalize this question using a bias-variance decomposition of the errors made by AI models: An AI's *error-incoherence* on a task is measured over test-time randomness as the fraction of its error that stems from variance rather than bias in task outcome. Across all tasks and frontier models we measure, the longer models spend reasoning and taking actions, *the more incoherent* their failures become. Error-incoherence changes with model scale in a way that is experiment dependent. However, in several settings, larger, more capable models are more incoherent than smaller models. Consequently, scale alone seems unlikely to eliminate error-incoherence. Instead, as more capable AIs pursue harder tasks, requiring more sequential action and thought, our results predict failures to be accompanied by more incoherent behavior. This suggests a future where AIs sometimes cause industrial accidents (due to unpredictable misbehavior), but are less likely to exhibit consistent pursuit of a misaligned goal. This increases the relative importance of alignment research targeting reward hacking or goal misspecification.

⌗ `hot-mess-of-ai`     🧑‍🦱 `hot-mess-data`

## 1 Introduction

There are an increasing number of predictions that AI will soon be more capable than human beings (Kwa et al., 2025; Maslej et al., 2025; Pimpale et al., 2025), and will replace human labor in many domains (Chen et al., 2025b; Handa et al., 2025; Dominski & Lee, 2025; Eloundou et al., 2024; Johnston & Makridis, 2025). We already rely on AI for consequential tasks such as writing critical software (DeepMind, 2025; Appel et al., 2025), determining bail amounts (Fine et al., 2025), and deciding what stories to present in news feeds (Liu et al., 2024; Gao et al., 2024b; Yada & Yamana, 2025). Despite its increasing capabilities, AI often behaves in ways we do not intend. Due to its high-stakes use cases, it is important to understand how and when AI can be expected to fail.

One class of AI risk is *misalignment risk* (Bostrom, 2014; Russell, 2019; Greenblatt et al., 2024). Misalignment risk is the concern that AI will pursue a goal that is different from the goal its creators intended to instill, and that it will pursue that goal with superhuman competence. If a superhuman agent pursues a misaligned goal, it might do things like seize power as an instrumental step to achieving its goal (Hubinger et al., 2019).

However, this scenario assumes that unintended behavior stems from systems that not only pursue the wrong objective, but remain coherent optimizers over a long horizon. Large language models (LLMs), prior to reinforcement learning, are dynamical systems, but not optimizers. They have to be trained to act as an optimizer, and trained to align with human intent. It is not clear which of these trained properties will tend to be more robust, and which will be most likely to cause failures

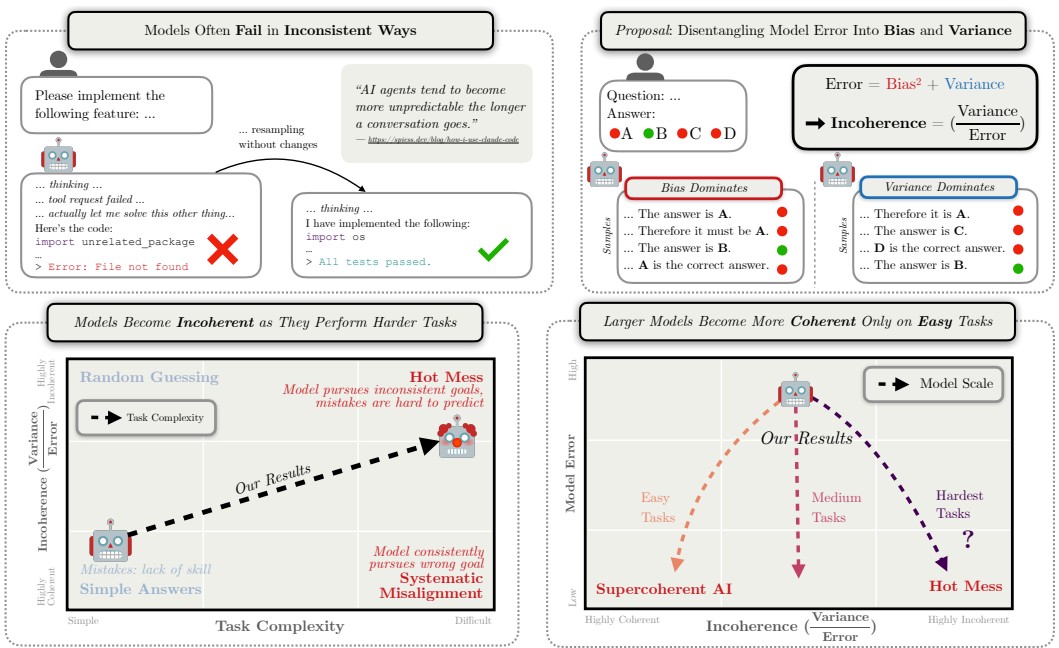

Figure 1: **AI can fail because it is misaligned, and produces consistent but undesired outcomes, or because it is incoherent, and does not produce consistent outcomes at all. These failures correspond to** *bias* **and** *variance* **respectively. As we extrapolate risks from AI, it is important to understand whether failures from more capable models performing more complex tasks will be bias or variance dominated. Bias dominated failures will look like model misalignment, while variance dominated failures will resemble industrial accidents.** (*top left*) Qualitatively, we observe that AI models fail in unpredictable and inconsistent ways. Often, these failures can be fixed by resampling. (*top right*) To quantify this observation, we decompose errors made by AI into two terms, bias and variance. We illustrate this using a multiple choice task: bias is the tendency to pick a specific incorrect answer; variance is the tendency to pick inconsistenly among options. We define error-incoherence as the fraction of model error caused by variance. (*lower left*) Experimentally, we find that as models reason longer and take more sequential actions, they become more incoherent. (*lower right*) We find that as models become more capable, and overall error rate drops, error-incoherence changes in a way that depends on task difficulty. Easy tasks become less incoherent, while hard tasks trend towards increasing error-incoherence.

in superhuman systems. In practice, AI models often fail in ways that seem random and do not further any coherent goal (Spiess, 2025; Nolan, 2025). Like humans, when AIs act undesirably, it is often because they are a *hot mess* and do not act in a way that is consistent with any goal: The *hot mess theory of intelligence* (Sohl-Dickstein, 2023) suggests that as entities become more intelligent, their behavior tends to become more incoherent, and less well described through a single goal. If true for AI systems, this shifts both the likelihood and the focus of misalignment scenarios.

In this paper, we therefore ask the questions: *When a model does something other than what we intend, what fraction of its deviation is due to* **bias** *(consistent pursuit of the wrong goal), and what fraction to* **variance** *(randomness in behavior and outcome)? As we scale model intelligence and task complexity, how does this decomposition change? Asymptotically, as extremely capable models perform extremely complex tasks, which class of undesired behavior will dominate?*

We address these questions by measuring the scaling behavior of AI errors decomposed into

$$\text{ERROR} = \text{BIAS}^2 + \text{VARIANCE} ,$$

and further define error-incoherence as the proportion of variance to the total error. This decomposition allows us to distinguish the *relative contributions* of different types of AI failure, and, importantly, how they change as models become more intelligent and perform longer horizon tasks. *Bias-dominated failures* correspond to systematic misalignment—consistent pursuit of the wrong objective—whereas *variance-dominated failures* indicate inconsistent outcomes.

We find that across multiple-choice benchmarks, agentic coding, and safety tasks, models become more incoherent with longer reasoning (Fig. 2), even when controlling for task difficulty (Fig. 3). Larger, more capable models are often more incoherent (Fig. 4): while they achieve lower error, they grow more coherent on easy tasks but less coherent on hard tasks (Fig. 5). We validate these findings in a synthetic environment where variance asymptotically dominates with increasing model size (Fig. 6), and find that ensembling and larger reasoning budgets reduce error-incoherence (Fig. 7). We discuss our results in Section 5.

## 2 BACKGROUND

### 2.1 BIAS–VARIANCE DECOMPOSITION

**Definition.** In supervised settings, the *bias–variance decomposition* expresses the expected error of a predictor as the sum of three terms: BIAS$^2$, VARIANCE, and irreducible noise (Kohavi & Wolpert, 1996). Although originally formulated for regression, analogous decompositions exist for classification tasks (Kohavi & Wolpert, 1996; Domingos, 2000), with a similar interpretation: the bias reflects the error of the classifier's **mean** or **mode** prediction and variance quantifies its deviation. Several such decompositions exist, including the $0/1$ error (Kong & Dietterich, 1995; Breiman, 1996; Kohavi & Wolpert, 1996; Tibshirani, 1996; Friedman, 1997; Domingos, 2000), Brier score (Degroot & Fienberg, 2018), and cross-entropy error (Heskes, 1998). We present a Kullback-Leibler (KL) decomposition in the main text. For additional definitions see Appx. A. We ran experiments with KL, Brier, and 0/1 formulations. All three decompositions produce qualitatively similar results, and we provide plots for all three in appendices.

Let $x$ be the input with label classes $c \in \{1, \ldots, C\}$ for which the model $f_\varepsilon$ produces a probability distribution (potentially one-hot) over class labels $f_\varepsilon(x) \in \mathbb{R}^C$, with $\varepsilon$ denoting the stochasticity of the training process. The target is one-hot encoded through $y(x) \in \mathbb{R}^C$. For clarity, we omit the dependence of $y$ and $f_\varepsilon$ on $x$. We assume the irreducible noise to be $0$. Then, the expected cross-entropy error can be decomposed into (Yang et al., 2020):

$$\underbrace{\mathbb{E}_\varepsilon\left[\mathrm{CE}(y, f_\varepsilon)\right]}_{\text{ERROR}} = \mathbb{E}_\varepsilon\left[\sum_{c=1}^{C} y[c] \log(f_\varepsilon[c])\right] = \underbrace{D_{\mathrm{KL}}\left(y \| \bar{f}\right)}_{\text{BIAS}^2} + \underbrace{\mathbb{E}_\varepsilon\left[D_{\mathrm{KL}}(\bar{f} \| f_\varepsilon)\right]}_{\text{VARIANCE}}, \tag{1}$$

where $y[c]$ denotes the $c$-th element of the vector, $D_{\mathrm{KL}}$ is the Kullback-Leibler divergence, and $\bar{f}_\varepsilon$ is the average of *log-probabilities* after normalization: $\bar{f}[c] \propto \exp\left(\mathbb{E}_\varepsilon\left[\log(f_\varepsilon[c])\right]\right)$ for $c = 1, \ldots, C$. We denote this decomposition as KL-BIAS and KL-VARIANCE. This is an instance of the general decomposition for Bregman Divergences (Pfau, 2013).

**Different usage to classical literature.** In discussions of the bias–variance tradeoff, the setup typically assumes a deterministic model (*e.g.,* a regressor), with bias and variance estimated by retraining under different seeds or data sampling. That means the expectation is over training randomness $\varepsilon$. Our setting differs: rather than retraining multiple models, we analyze a *fixed model* and take the expectation over input (*e.g.,* few-shots) and output (sampling) randomness $\varepsilon$ for *the same task*.

**Error-incoherence.** Throughout this paper, our main metric of interest is the *proportion of the variance to the total error*, which we define as ERROR-INCOHERENCE. Formally, consider a set of questions $Q = \{q_i\}_{i \leq N}$ and a model $f_\varepsilon$. We then denote error-incoherence as

$$\text{ERROR-INCOHERENCE}(Q, f_\varepsilon) := \frac{\sum_i \text{VARIANCE}(q_i, f_\varepsilon)}{\sum_i \text{ERROR}(q_i, f_\varepsilon)}. \tag{2}$$

Since $\text{ERROR}(q_i, f_\varepsilon) = \text{BIAS}(q_i, f_\varepsilon)^2 + \text{VARIANCE}(q_i, f_\varepsilon)$, this metric is a *relative* value in $[0, 1]$: a value of $0$ means that the model never deviates from its average behavior and any error will be consistent; a value of $1$ means that every error the model makes is inconsistent. Importantly, a model can achieve a lower overall error rate, but have a higher error-incoherence, which makes it a comparable measure across error levels and model capabilities. We see such cases in Section 3.

### 2.2 SCALING BEHAVIOR OF LARGE LANGUAGE MODELS

**Scaling laws.** Model performance generally follows predictable *power-law scaling* with respect to model size $N$, dataset size $D$, and compute $C$ (Kaplan et al., 2020; Hoffmann et al., 2022). Most

prominently, taking the parameters $N$ as an argument, the cross-entropy loss broadly behaves as $l(N) \propto N^{-\alpha}$ for some exponent $\alpha$. This slope $\alpha$ informs us about the *rate* of improvement. In Section 3.2 we will compute scaling laws independently for bias and variance loss contributions, to judge which asymptotically dominates.

**Reasoning and inference compute.** Besides the model and dataset size, the most promising recent development uses *inference compute* as an axis of scale. Specifically, so-called reasoning models are trained with reinforcement learning (RL) to think in long chains of thought before providing an answer, which improves performance with larger thinking budgets (Snell et al., 2025; Jaech et al., 2024; Guo et al., 2025; Anthropic, 2025b; OpenAI, 2025a; Team, 2025a; Team et al., 2025; Chen et al., 2025a; Zhong et al., 2024; Muennighoff et al., 2025). The length of reasoning is an important aspect of our analysis, which we see as a process of sequential action steps (Lightman et al., 2023).

## 3 EXPERIMENTS

**Overview.** We present our results grouped by observations: first, growing incoherence as a function of reasoning length (3.1) and scaling laws with model scale (3.2); this is followed by the effects of reasoning budgets and ensembling (3.3). The details of all experimental setups are in Appx. B.

**Tasks.** We run experiments on the following tasks, which all have well-defined targets used for incoherence measurements, since bias is only defined relative to a target. For a discussion, see Section 5.

- **Multiple Choice Tasks.** We use the popular scientific reasoning benchmark GPQA (Rein et al., 2024), and general knowledge benchmark MMLU (Hendrycks et al., 2021). Target responses are simply the correct answer.
- **Agentic Coding.** This focuses on SWE-BENCH (Jimenez et al., 2024), where agents solve GitHub issues using tools, and success is measured with unit tests.
- **Safety and Alignment.** We assess models using the advanced AI risk subset of Model-Written Evals (MWE; Perez et al., 2023), both with the original multiple choices and in an open-ended format with answer options removed.
- **Synthetic Settings.** We train transformers of varying scales to directly emulate an optimizer descending an ill-conditioned quadratic loss. The transformer is tasked with predicting string representations of optimizer update steps based on the current state. This is a simple toy model of an LLM that has been trained to act as an optimizer. See Section 3.2.2 for details.
- **Survey.** In addition to experiments using LLMs, we report the survey results of Sohl-Dickstein (2023) (previously released in blog form), where disjoint sets of human subjects subjectively ranked the intelligence and coherence of AI models, humans, non-human beings, and organizations. The details are provided in Appx. B.5.

**Setup and Metrics.** Across all tasks, unless otherwise noted, we obtain at least 30 samples to estimate bias and variance per question. We find this sample count to be sufficient for stable estimates (see Appx. C.5 and B). Each sample is run with a different seed for autoregressive generation. For GPQA and MMLU, samples additionally use a different random few-shot context. We report the following metrics (details in Appx. A and B):

- For multiple choice questions, our main metric of interest is the KL-ERROR-INCOHERENCE, *i.e.,* the error-incoherence with respect to KL-BIAS and KL-VARIANCE (Equations 1 and 2). We find the same qualitative behavior for other decompositions, as reported in Appx. C.1.
- For open-ended MWE safety questions, we embed solely the answers (*i.e.,* without reasoning chains) using a text embedding model (text-embedding-3-large). Consequently, we report the *variance of the embedding vectors* in the Euclidean norm.
- For SWE-BENCH, we assign binary vectors for each sample and task: each vector is of size $T_i$, the number of unit tests for task $i$, and encodes which tests a model's code passes. The *coverage error* then computes the mean squared difference to a vector of all 1's, which we decompose into bias and variance contributions.

**Models.** We evaluate the following frontier models: SONNET 4 (Anthropic, 2025a) with reasoning enabled, O3-MINI (OpenAI, 2025a), and O4-MINI (OpenAI, 2025b). When analyzing scaling *w.r.t.* model size as an imperfect proxy for intelligence, we use the QWEN3 model family with thinking

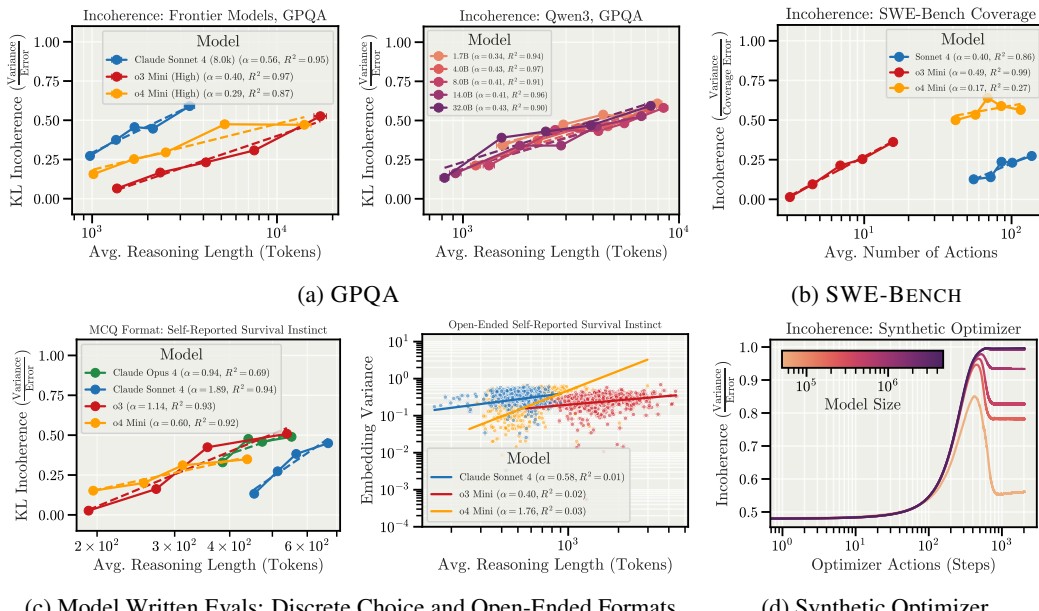

(a) GPQA

(b) SWE-BENCH

(c) Model Written Evals: Discrete Choice and Open-Ended Formats

(d) Synthetic Optimizer

Figure 2: **Across a variety of settings, as models reason longer or take more actions, they become more incoherent.** We assess frontier models (SONNET 4, O3-MINI, O4-MINI, QWEN3) across a variety of different tasks (MCQ, Agentic Coding, Alignment). We evaluate with *many samples* to estimate bias and variance terms for each question. When sorting questions by average reasoning lengths and grouping into buckets, a clear trend emerges: error-incoherence increases significantly with reasoning length. In other words, for questions where models reason longer and take many actions, their errors are dominated by variance. We make a similar observation for the variance of text embeddings to open-ended safety questions *((c), right)*, and in a synthetic setting *(d)*.

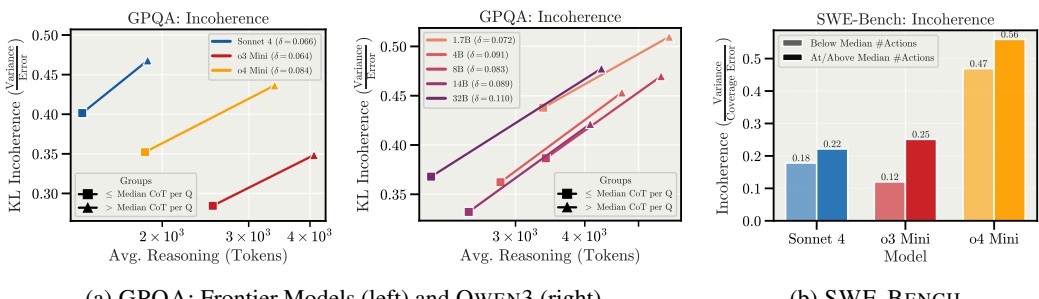

(a) GPQA: Frontier Models (left) and QWEN3 (right)

(b) SWE-BENCH

Figure 3: **For a fixed task and reasoning budget, natural variation in reasoning length and actions is predictive of error-incoherence.** We analyze GPQA (left, *(a)*) and SWE-BENCH *(b)* by splitting samples into above- or below-median reasoning length (GPQA) or actions (SWE-BENCH) *per question*. We then compute performance and incoherence for both groups. *(a)* The naturally longer reasoning shows increased incoherence for both frontier models (left) and QWEN3 (right). *(b)* Similar observations apply to SWE-BENCH, where longer action sequences display higher incoherence for test coverage (right). This effect is much stronger than through larger reasoning budgets (Fig. 7), and the difference in accuracy or score is minimal between both groups (Fig. 17).

enabled (Team, 2025a). In Sect. 3.2.2, we train our own autoregressive transformers on a synthetic optimization task.

## 3.1 THE RELATION BETWEEN REASONING LENGTH, ACTION LENGTH AND INCOHERENCE

*The longer models spend reasoning and taking actions, the more incoherent they become.*

**Sorting by reasoning & action length.** We begin with a key experimental observation. Fig. 2 shows all setups with reasoning tokens (or actions for SWE-BENCH, optimization steps for the

synthetic setting) on the x-axis and error-incoherence or variance on the y-axis. For Figures 2(a) to 2(c), lines show different question sets across and within models, obtained by sorting by average length and grouping into equal buckets, with error-incoherence computed per group.

Across all conditions, longer reasoning and action sequences increase error-incoherence. For GPQA, it increases with different slopes per model family (and reasoning length distributions); notably, for QWEN3, incoherence levels and slopes are nearly identical across all sizes, even though larger models perform better (cf. Figure 9). Similar patterns appear for frontier models on MWE. For SWE-BENCH, both baseline error-incoherence and slopes vary: O4-MINI shows higher baseline incoherence but smaller slope; O3-MINI has the largest slope but lowest baseline incoherence.

**Example analysis.** To illustrate, we provide real experimental transcripts in Fig. 19. The example shows SONNET 4 responding differently with nearly every sample to a disconnection question, displaying high error-incoherence. This connects to open-ended MWE results in Fig. 2(c), where embedding variance correlates strongly with average reasoning length, and bias is not well-defined. We provide additional insight on error-incoherence through absolute answer change rates in Appx. C.4, and all open-ended MWE plots in Fig. 24.

**Discussion: Task complexity.** Sorting questions by reasoning length implicitly selects for *task difficulty* (see accuracies in Fig. 8 and 9), suggesting error-incoherence is higher when making mistakes on more complex tasks. While perhaps unsurprising, this is an important experimental observation. In fact, for frontier models, our setup asks models for probability estimates of choice correctness (see Appx. B.1), *i.e.,* we give them an option to express uncertainty. We revisit task complexity in the next section and Section 3.3.

**Natural overthinking and error-incoherence.** Irrespective of task complexity, we show how long reasoning and action sequences lead to larger error-incoherence in Fig. 3. For each question, we assign response samples to either of two groups: those below and those above the median reasoning length for this specific question for GPQA, and the median number of actions for this task in SWE-BENCH. The incoherence is substantially higher for the second group for both benchmarks. Notably, the average accuracy and SWE-BENCH-score (shown in Fig. 17) is similar between groups, but the effect of the natural variation on error-incoherence is much larger than reasoning budgets (Fig. 7(a)).

**Further results.** We provide more analyses for GPQA in Appx. C.1, with reasoning length correlations in Appx. C.6. Results for MWE are in Appx. C.7, and results for SWE-BENCH in Appx. C.8.

## 3.2 THE RELATION BETWEEN MODEL SCALE, INTELLIGENCE, AND ERROR-INCOHERENCE

*Larger and more intelligent systems are sometimes more incoherent.*

**Motivation.** In Section 3.1, in particular Fig. 2(a), we fix a model and analyze error-incoherence as a function of reasoning length. Now, we ask a different question: *When we fix a task, how does error-incoherence change as a function of model size? How does incoherence scale with intelligence?*

**Overview.** We summarize the main observation in Fig. 4: larger, more capable and intelligent systems are often more incoherent. This is manifested in LLMs for the most complex set of questions (Sect. 3.2.1), the rankings of intelligence and error-incoherence as judged by human survey participants (Appx. B.5) and our synthetic optimizer setting (Sect. 3.2.2). However, we find that larger models are less incoherent on simpler questions (Sect. 3.2.1). We discuss each result in detail.

### 3.2.1 SCALING LAWS FOR LLMS SEPARATED BY TASK COMPLEXITY

*Easy tasks become less incoherent with scale, while harder tasks become more incoherent.*

**Overview.** We experiment with the QWEN3 model family, as they provide the same model architecture, including reasoning abilities, with up to 32B parameters. Consistent with other setups, we sample many responses for the same set of questions. Additionally, we cluster questions using the the reasoning length of a reference model (here: 32B) into equally sized groups.

**Results.** See Fig. 5 for the detailed results. We find that performance consistently improves with increasing model size, with the fastest rate of improvement for the hardest questions. However, the way in which error-incoherence changes with scale depends on question difficulty: Model responses to easy questions become more coherent with scale, while responses to the hardest questions become more incoherent with scale, though this last trend is noisy.

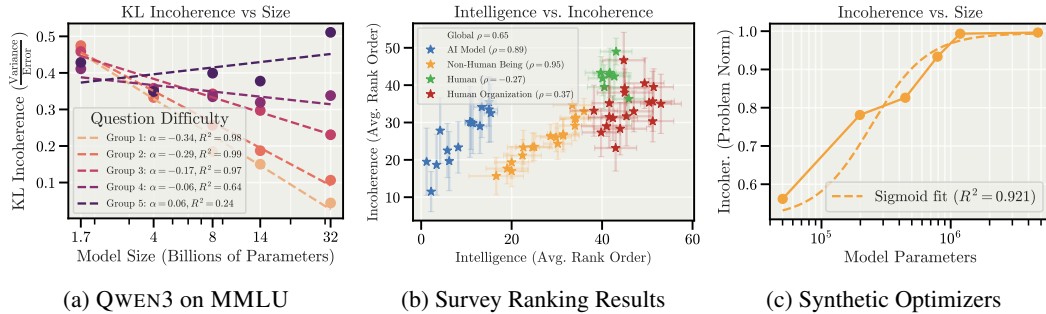

(a) QWEN3 on MMLU  (b) Survey Ranking Results  (c) Synthetic Optimizers

Figure 4: **Larger and more intelligent systems are often more incoherent.** *(a)* We measure the scaling of error-incoherence vs. model size for the QWEN3 family, as a function of question difficulty on MMLU. For easy questions, error-incoherence drops with model scale, while for the hardest questions error-incoherence remains constant or increases with model scale. The expanded results for this experiment are in Fig. 5. *(b)* Disjoint sets of human subjects were tasked with subjectively ranking the intelligence and incoherence of diverse AI models, non-human beings, well known humans, and human organizations. Across all categories, entities that were judged more intelligent by one group of subjects, were independently judged to be more incoherent by another group of subjects. See Appx. B.5. *(c)* In a synthetic task, we train transformers of increasing size to explicitly emulate optimizer trajectories descending a quadratic loss. As these models become larger, the trajectories they generate achieve lower loss on the quadratic. However, the final loss is also more variance dominated and thus incoherent with increasing model size. Details in Fig. 6.

**Further results.** We provide different visualizations of the same results in Appx. C.2, which include the same results for GPQA (Fig. 12), the relationship between error-incoherence and error (Fig. 13) and how reasoning length is a stronger indicator of error-incoherence than model size (Fig. 14).

### 3.2.2 SCALING LAWS IN CONTROLLED SYNTHETIC SETTINGS: MODELS AS OPTIMIZERS

*On a synthetic task, models become more incoherent as they are made larger.*

**Models as optimizers.** In this paper, we are trying to disentangle whether capable models will more tend to act as effective optimizers of the wrong goal, or will pursue the right goal but not be effective optimizers. To quantify this in a controlled setting, we train models to literally mimic the trajectory of a hand-coded optimizer descending a loss function. This can be viewed as trying to train a model to implement a mesa-optimizers (Hubinger et al., 2019). We then analyze the bias and variance of the resulting models, to answer the question: *Does the model become an optimizer faster or slower than it converges on the right optimization objective?*

**Setup.** We study a simple $d$-dimensional quadratic function of the form $f(x) = \frac{1}{2}(x-b)^T A(x-b)$, where $A \in \mathbb{R}^{d \times d}$ is a (random) positive-definite but ill-conditioned matrix. We set the condition number to $50$. Training data is generated by using an optimizer to produce many trajectories of fixed length for random initial points. The optimizer used to generate the training data performs steepest descent with a fixed step norm. The training dataset consists of pairs $(x_i, u_i)$, where $x_i$ is a parameter iterate, and $u_i$ is the corresponding update step generated by the optimizer. Analogously to real (token-based) models, we train transformer models (Vaswani et al., 2017) of varying sizes using *decoding-based regression* (Song & Bahri, 2025) and teacher forcing. This means we tokenize the scientific format representation of $x_i$ and $u_i$, with a vocabulary of digits and signs. When evaluating, we sample multiple initial points and roll out trajectories using the model's own predictions. A visualization of this with a real model is provided in Fig. 6 (left). The bias and variance measures are then taken *w.r.t.* the optimum and norm $\|\cdot\|_A$ that is induced by the problem. The details are in Appx. B.4.

**Results.** The main results are shown in Fig. 2(d) (error-incoherence over rollout steps) and Fig. 6 (scaling laws by size). All models show consistently rising error-incoherence per step; interestingly, smaller models reach a lower plateau after a tipping point where they can no longer follow the correct trajectory and stagnate, reducing variance. This pattern also appears in individual bias and variance curves (Fig. 26). Importantly, larger models reduce bias more than variance. These results suggest that they learn the correct objective faster than the ability to maintain long coherent action sequences. More results and discussions are provided in Appx. C.9.

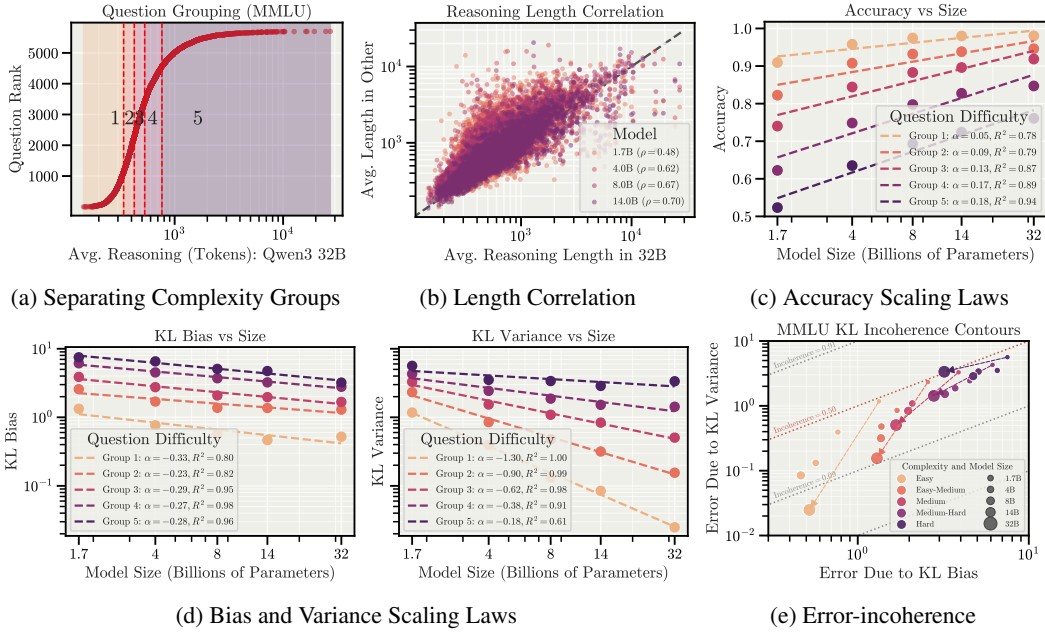

Figure 5: **Details for QWEN3 scaling laws: easy tasks become less incoherent, harder tasks more incoherent.** We group MMLU questions by reasoning length using a reference model (Qwen3 32B, *(a)*), which correlates across model sizes *(b)* and serves as a task complexity proxy, as accuracy drops with longer reasoning *(c)*. These groups reveal distinct bias–variance scaling *(d)*: bias slopes are similar across groups, but variance slopes decrease sharply for harder ones. In the hardest group, variance slopes fall below bias slopes, leaving variance as the limiting factor. Thus, larger models remain constrained by variance and *more incoherent with scale (e)*. We provide more analyses including other models and the same conclusion for GPQA in Appx. C.2.

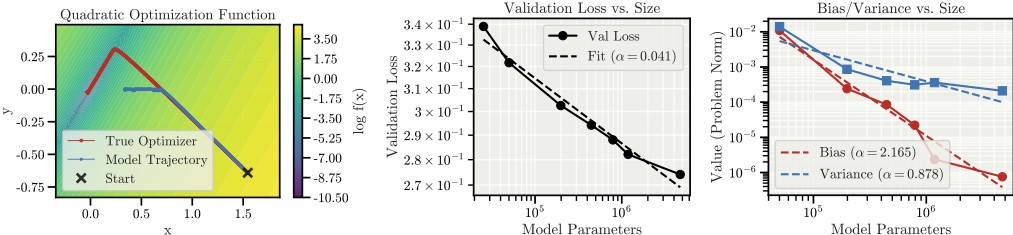

Figure 6: **Details for synthetic optimization: In controlled settings with teacher forcing and a single objective, language models become variance dominated with increasing size.** (*left*) We train autoregressive transformers to predict update steps to minimize a quadratic function using decoding based regression, *i.e.,* next-token prediction. This setting involves sequentially performing steps towards a goal via next token prediction, emulating a key feature of goal seeking AI. (*middle*) The loss (next-token prediction objective) follows a clear power law improvement with model size. (*right*) When evaluating the trained models using their own rollouts, we find that increasing model size reduces bias much faster than variance.

## 3.3    THE EFFECTS OF REASONING BUDGET AND ENSEMBLING

We now study the effect of reasoning budgets, *i.e.,* the techniques provided in model APIs, and ensembling, *i.e.,* averaging multiple responses, on error-incoherence. The main results are in Fig. 7.

### 3.3.1    REASONING BUDGETS

*Reasoning budgets reduce error-incoherence, but natural variation has a much stronger effect.*

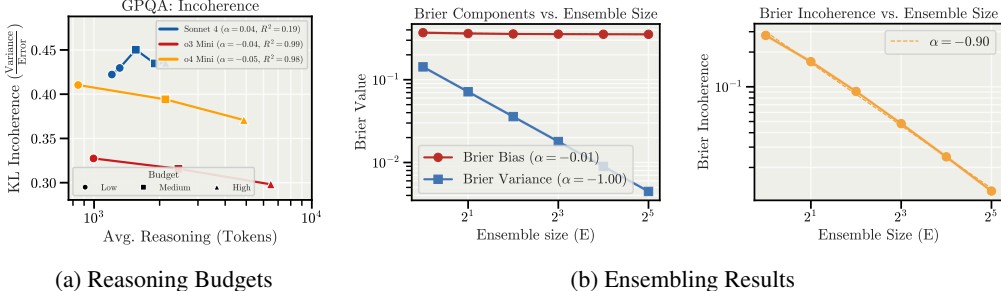

(a) Reasoning Budgets            (b) Ensembling Results

Figure 7: **Ensembling and larger reasoning budgets reduce error-incoherence. Other forms of error correction may also reduce error-incoherence.** *(a)* Instructing models to reason longer improves performance (inference scaling laws, Fig. 17) and sometimes error-incoherence. This effect is smaller than natural variation, where error-incoherence rises sharply (Fig. 3; direct comparison in Fig. 17). *(b)* With O4-MINI on GPQA, we analyze the effect of the *ensembling*, *i.e.,* using multiple samples to average output probabilities over targets for the same question. The bias and variance are now computed by comparing different ensembles of the same size. We find that, as expected from theory, it reduces variance with a rate of $1/E$, without affecting bias (*left*). As a consequence, error-incoherence drops (*right*). Ensembling is a particular form of model error correction, which is impractical for action loops in the world, since state can typically not be reset. However, we expect other error correction techniques to also reduce incoherence.

**Inference scaling.** We show the results of our inference-scaling analysis on GPQA in Fig. 7(a) and Fig. 17. Increasing reasoning budgets improves performance (17(a), left), and slightly reduces error-incoherence for all models but SONNET 4 (7(a)). Interestingly, this effect is overshadowed by error-incoherence that arises through natural variation, *i.e.,* when models think longer than the median for a question (recall analysis in Fig. 3; direct comparison in Fig. 17(a), right).

**Discussion: How does reasoning budget improve coherence?** Since the implementation details of reasoning budgets for frontier models are not public, it is unclear how exactly it can improve error-incoherence. We believe it is likely explained by better backtracking and error correction properties, a phenomena observed to arise during training with larger budgets (Guo et al., 2025), and related to the ensembling results in Sec. 3.3.2. We partially explore error-incoherence through the reasoning structure with the QWEN3 reasoning traces in Appx. C.3.

### 3.3.2 ENSEMBLING

*Ensembling multiple attempts reduces error-incoherence.*

**Motivation.** Perhaps the most natural way to reduce error-incoherence is to ensemble multiple attempts: instead of relying on a single answer, we roll out multiple trajectories and combine them. We demonstrate this with a repetition of the experiment for GPQA with O4-MINI.

**Setup.** We obtain 320 samples of answers for all questions of GPQA. Fixing an ensemble of size $E$, we average the $E$ produced probabilities over targets. To compute bias and variance, we then compare ensembles of the same size across random samples of ensembles, which we hold at a fixed number of 10, while ensuring that samples do not overlap. This allows ensemble sizes of up to 32.

**Results.** Fig. 7(b) shows how variance changes with increasing ensemble size. As expected, it drops like the inverse of the ensemble size, and error-incoherence therefore also drops. We expect there are broader classes of error correction that behave similarly. The slight reduction in error-incoherence with increasing reasoning budgets in Sec.3.3.1 may be achieved through such a mechanism. We provide the plots for KL-ERROR-INCOHERENCE in Fig. 11.

## 4 RELATED WORK

We summarize the most important related work and defer a comprehensive discussion to Appx. D.

**Reasoning.** Recent studies report inverse scaling trends with extended reasoning degrading performance (Gema et al., 2025; Su et al., 2025; Wu et al., 2025; Hassid et al., 2025). Most relevant,

Ghosal et al. (2025) find that overthinking increases output variance, though via artificially injected tokens rather than natural overthinking. While these studies identify performance degradation, they do not distinguish systematic errors from inconsistent failures. Our ensembling analysis relates to self-consistency work (Wang et al., 2023), but reframes aggregation as reducing error-incoherence.

**Evaluation variance.** Even though AI models have vastly improved upon benchmarks, evaluations are known to be highly variant (Bui et al., 2025; Biderman et al., 2024). Errica et al. (2025) formalize this through sensitivity and consistency metrics, revealing important failure modes. This is similar setup to our input and output randomness. Importantly, we connect the variability to the concepts of bias and variance, highlighting the relevance in the safety setting, and analyze scaling laws.

**Scaling behavior.** As models get larger and more capable, evidence suggests their representation and errors become highly aligned (Kim et al., 2025; Huh et al., 2024; Goel et al., 2025) and that they improve long-horizon tasks (Sinha et al., 2025). Our work complements these observations by finding increased error-incoherence the longer models reason and act, aligned between model families.

## 5 DISCUSSION AND WHAT OUR RESULTS DO NOT TELL US

**Why expect more capable models to be more incoherent?** In this paper, we do not experimentally or theoretically explore the specific mechanisms for increasing error-incoherence with increasing trajectory length and (sometimes) model size. However, there are motivating observations.

The first is that LLMs are dynamical systems. When they generate text or take actions, they trace trajectories in a high-dimensional state space. It is often *very hard* to constrain a generic dynamical system to act as an optimizer. The set of dynamical systems that act as optimizers of a fixed loss is measure zero in the space of all dynamical systems. As models scale and acquire broader capabilities, their effective state and action space expands, exacerbating this difficulty. We should not expect AIs to act as optimizers without considerable effort, nor should we expect this to be easier than training other properties into their dynamics.

Second, variance typically accumulates over a trajectory unless there is an active correction mechanism (like ensembling, Fig. 7). When an AI acts in the real world, actions are often irreversible. Therefore, it will often be impossible or impractical to correct for noise introduced by model actions.

**Reward misspecification.** Bias can be further decomposed into $\text{BIAS} = \text{BIAS}_{\text{MESA}} + \text{BIAS}_{\text{SPEC}}$, where $\text{BIAS}_{\text{MESA}}$ captures the average deviation of the model's behavior from the training objective, and $\text{BIAS}_{\text{SPEC}}$ captures the deviation of the training objective from the *intended* training objective. For our tasks, we believe that there was not meaningful reward misspecification. In settings with poorly specified training objectives, we worry that $\text{BIAS}_{\text{SPEC}}$ would come to dominate the error, as both variance and $\text{BIAS}_{\text{MESA}}$ go to zero with increasing model capability. Our results underscore the importance of characterizing and mitigating goal misspecification during training.

**Open-ended goals and error-incoherence.** To rigorously analyze the scaling of bias, variance, and error-incoherence, we need to (1) measure an "average" prediction (for bias and variance) and (2) measure distance to ground truth (for bias). We use multiple-choice classification, coding unit-tests, and objective functions rather than LLM judges to ensure metrics are well-defined, unbiased, and comparable. Extracting hidden goals and complex incoherent behaviors remains important (cf. Section 4.1.1.5; Anthropic, 2025a); our embedding-variance analysis of model-written evals (Appx.C.7) provides an initial exploration of a setting where bias is not easily defined or measured.

## 6 CONCLUSION

Motivated by the hot mess theory of AI misalignment, we propose a bias–variance decomposition as a framework for analyzing how increasingly capable AIs will fail. Our results show that longer sequences of reasoning and actions consistently increase model error-incoherence. We also find that smarter AI models are not consistently more coherent. Our results suggest that when advanced AI systems performing complex tasks fail, it is likely to be in inconsistent ways that do not correspond to pursuit of any stable goal. This should inform judgements of the relative plausibility of different AI risk scenarios and guide further research into understanding the mechanistic origins of error-incoherence.

ACKNOWLEDGEMENTS

We thank Andrew Saxe, Brian Cheung, Kit Frasier-Taliente, Igor Shilov, Stewart Slocum, Aidan Ewart, David Duvenaud, and Tom Adamczewski for extremely helpful discussions on topics and results in this paper.

ETHICS STATEMENT

This research aims to characterize failure modes of increasingly capable AI systems to inform safer deployment strategies. Our findings suggest that as AI systems tackle more complex tasks requiring extended reasoning, incoherent failures become more prevalent than systematic misalignment. While this work does not directly prevent AI failures, it offers empirical grounding for prioritizing safety interventions, suggesting greater focus on preventing unpredictable accidents rather than solely defending against coherent malicious behavior. We believe this understanding of AI failure modes benefits the community to ensure safe AI deployment.

REPRODUCIBILITY STATEMENT

We provide a detailed description of our theoretical framework in Section 2.1 and Appx. A. The general experimental setups are described in Section 3 and Appx. B, with task-specific details outlined in each experiment subsections. Our code and data is available here.

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

CONTENTS

## A   BIAS AND VARIANCE DEFINITIONS FOR CLASSIFICATION

Recall the classical bias-variance decompositon in the case of regression: Considering the mean-squared error for a sample point $(x, y) \in \mathbb{R}^2$, the decomposition is given by

$$\text{MSE} = \mathbb{E}_\varepsilon[(y - f_\varepsilon(x))^2] = \underbrace{(\mathbb{E}_\varepsilon[f_\varepsilon(x)] - f(x))^2}_{\text{BIAS}^2} + \underbrace{\mathbb{E}_\varepsilon[(f_\varepsilon(x) - \mathbb{E}_\varepsilon[f_\varepsilon(x)])^2]}_{\text{VARIANCE}} + \underbrace{\sigma^2}_{\text{Irreducible Error}}, \quad (3)$$

where $f$ is the ground-truth function, and the expectation is taken w.r.t. the randomness $\varepsilon$ in the training process (*e.g.,* data ordering) that the model $f_\varepsilon$ depends on.

**Classification Formulation.** While the interpretation for classification is similar, different decompositions exist, which we review in the following. Throughout this section, let $x$ be the input of a problem with target class $c(x) \in \{1, \ldots, C\}$ and one-hot target $y(x) \in \mathbb{R}^C$. The model $f_\varepsilon$ produces a probability distribution (potentially one-hot) over class labels $f_\varepsilon(x) \in \mathbb{R}^C$. For clarity, we omit the dependence of $c$, $y$ and $f_\varepsilon$ on $x$. $y[c]$ denotes the $c$-th element of the vector. Throughout our experiments and derivations, we assume that the irreducible noise is 0 (*i.e.,* no stochasticity in the data-generating process or wrong labels) for simplicity. Note that each of the following decompositions gives bias and variance for a single data point $(x, y)$, which is aggregated over a dataset $\{(x_i, c_i)\}_i$.

**0/1 Error.**   The classical decomposition for a 0/1 loss relies on the unified decomposition by Domingos (2000). Let $c(x)$ be the ground-truth class (assuming noiseless labelling) and the model's predicted class be $c_\varepsilon(x) = \arg\max_c f_\varepsilon(x)[c]$. The *systematic* mean is $\bar{c} = \arg\max_c \mathbb{E}_\varepsilon[f_\varepsilon[c]]$, *i.e.,* the mode of the average prediction. Then, the 0/1 loss $L$ for sample $x$ can be decomposed into

$$\mathbb{E}_\varepsilon[L(c, c_\varepsilon)] = \mathbb{E}_\varepsilon[\mathbf{1}\{c \neq c_\varepsilon\}] = \underbrace{\mathbf{1}\{c \neq \bar{c}\}}_{\text{BIAS}^2} + a \cdot \underbrace{\mathbb{E}_\varepsilon[\mathbf{1}\{\bar{c} \neq c_\varepsilon\}]}_{\text{VARIANCE}}, \quad (4)$$

where the variable $a \in \{-1, 1\}$ is a multiplicative factor that enables the decomposition with a positive variance. In this setting, the bias is always either 0 or 1, and the variance captures the probability of deviating from the mode. Though universal, this decomposition has one major drawback: when computing an average over a dataset of questions $(x_i, c_i)_i$, it does not allow to average the bias and variance terms separately; instead, the decomposition only holds with the aforementioned multiplicative factor $a_i$. Formally, we have

$$\mathbb{E}_{(x_i,c_i),\varepsilon}[L(c_i, c_\varepsilon)] = \mathbb{E}_{(x_i,c_i),\varepsilon}[a_i \cdot \text{VARIANCE}_i] + \mathbb{E}_{(x_i,c_i),\varepsilon}[\text{BIAS}^2{}_i]$$
$$\neq \mathbb{E}_{(x_i,c_i),\varepsilon}[\text{VARIANCE}_i] + \mathbb{E}_{(x_i,c_i),\varepsilon}[\text{BIAS}^2{}_i];.$$

Essentially, the factor $a_i$ depends on the mode prediction being correct or not. We therefore report absolute bias and variance errors for the 0/1 loss in the Appendix, but do not compute error-incoherence.

**Brier Score.**   Similar to regression, we can treat the model's probability predictions as $C$-dimensional vectors to compute the mean square errors. Formally, the Brier score for multiclass prediction is defined and can be decomposed as

$$\mathbb{E}_\varepsilon[\text{BRIER}(y, f_\varepsilon)] = \mathbb{E}_\varepsilon[\|y - f_\varepsilon\|_2^2] = \mathbb{E}_\varepsilon\left[\sum_{c=1}^C (y[c] - f_\varepsilon[c])^2\right] = \underbrace{\|y - \hat{f}\|_2^2}_{\text{BRIER BIAS}^2} + \underbrace{\mathbb{E}_\varepsilon\left[\|\hat{f} - f_\varepsilon\|_2^2\right]}_{\text{BRIER VARIANCE}},$$

where $\hat{f} = \mathbb{E}_\varepsilon[f_\varepsilon]$ is the average prediction.

**KL Divergence (Cross-Entropy).**   The expected cross-entropy loss can be decomposed into

$$\mathbb{E}_\varepsilon[\text{CE}(y, f_\varepsilon)] = \mathbb{E}_\varepsilon\left[\sum_{c=1}^C y[c]\log(f_\varepsilon[c])\right]$$
$$= \underbrace{D_{\text{KL}}(y\|\bar{f})}_{\text{KL-BIAS}} + \underbrace{\mathbb{E}_\varepsilon[D_{\text{KL}}(\bar{f}\|f_\varepsilon)]}_{\text{KL-VARIANCE}}, \quad (5)$$

where $D_{\text{KL}}$ is the Kullback-Leibler divergence and $\bar{f}$ is the average of *log-probabilities after normalization*, *i.e.,*

$$\bar{f}_\varepsilon[c] \propto \exp(\mathbb{E}_\varepsilon[\log(f_\varepsilon[c])]) \text{ for } c = 1, \ldots, C.$$

Note that this is not the standard average prediction, as is the case in the Brier decomposition, but a geometric mean. In practice, since predicted probabilities can be zero, we apply Laplace smoothing to avoid $\log(0)$ or infinite values. This is done by updating the probabilities to $\hat{f}_\varepsilon[c] = \frac{f_\varepsilon[c]+\delta}{1+C\cdot\delta}$ for each $c = 1, \ldots, C$ with a small value of $\delta = 10^{-12}$.

# B EXPERIMENTAL DETAILS

## B.1 GPQA AND MMLU

**Setup.** We rely on the LM Harness (Gao et al., 2024a) codebase, where we evaluate models in multiple choice formats with custom written answer extraction functions to avoid false positives and negatives. For frontier models, we use reasoning budgets provided by the API (`low, medium, high` for the o-series, 1024-16k for Anthropic), with a maximum generation length of 32k for SONNET 4 and 100k tokens for the o-series. For QWEN3, we perform inference with vllm (Kwon et al., 2023) and recommended parameters for thinking (temperature 0.6, top-k 20, top-p 0.95). Since we consider multiple choice questions that only require a letter to answer, we count reasoning length using the amount of output tokens in the answer, either by the API count or using the actual tokenizer of QWEN3. To estimate the bias and variance metrics across both input (context) and output (sampling) randomness, we evaluate models using 10 different few-shot contexts randomly sampled from the corpus, and 3 samples for each fixed few-shot per question. This results in 30 samples per question overall. For MMLU, to reduce computational complexity, we limit 100 samples per question category (5700 in total).

**Probability prompting.** To provide models the option to express uncertainty and therefore reduce error-incoherence, we evaluate frontier models separate setup in addition to standard multiple-choice. We use the following prompt to ask for a probability estimate of each answer choice being correct:

> **Probability Format for MCQ**
> You will answer multiple-choice questions. Each question has a single correct answer. Work through each problem step-by-step, showing your reasoning and applying relevant concepts. Instead of choosing a single answer, YOU MUST PROVIDE an estimate of the probability of each answer being correct within "<PROB>P(A), P(B), P(C), P(D)</PROB>", where each P(X) is a float value between 0 and 1. The probabilities must sum to 1: P(A) + P(B) + P(C) + P(D) = 1. For example, if you think the probability of answer (A) being correct is 0.5, the probability of answer (B) being correct is 0.2, the probability of answer (C) being correct is 0.2, and the probability of answer (D) being correct is 0.1, then your response must end with "<PROB>0.5, 0.2, 0.2, 0.1</PROB>". You MUST end your response with the probabilities in the format specified above.

We report results for both standard and probabilty prompting in Appx. C.1, which show qualitatively the same behavior and performance. Frontier models are able to adhere to the format well, with only a few outliers (Table 1). Our main text shows the results for the probability format.

## B.2 MODEL-WRITTEN EVAL

We evaluate the models using the advanced AI risk evaluation subset from Perez et al. (2023). These tasks assess LLMs' self-reported behaviors relevant to advanced AI safety, including self-preservation inclinations, willingness to accept modifications to training objectives, and related safety-critical behaviors. We specifically use the human-generated subset to ensure higher evaluation quality.

**Setup.** Our experimental setup builds upon the codebase from Gema et al. (2025), which uses the *safety-tooling* library (Hughes & safety research, 2025) for API model inference. We conduct experiments under two conditions: the original multiple-choice format, and an open-ended format where we remove the multiple-choice options from the original questions. For both conditions, we compute the bias-variance decomposition with respect to the percentage of responses that align with desired safety properties. To ensure consistent evaluation across both formats, we employ the same system prompt that facilitates straightforward extraction of the model's final answer:

> **System Prompt for Model-Written Evals**
> Use a thinking process to analyze the problem step-by-step. At the end, provide your answer and clearly indicate it using <answer>X</answer> format.

In both cases, we obtain exactly 30 samples by simply resampling from the APIs. We use the returned output token count as a measure of reasoning length.

**Embeddings.** For the open-ended question set, we extract the model answers inside `<answer>` tags (*i.e.,* removing chain of thought or reasoning) and embed the text into fixed-size vectors using the OpenAI text embedding model `text-embedding-3-large`[1]. For the 30 samples per question, we in turn compute the variance in Euclidean space by computing the mean embedding and computing the average squared distance of samples to the mean.

### B.3 SWE-BENCH

**Setup.** We employ the Inspect Evals library (AI Security Institute, 2024) to evaluate models on SWE-BENCH (Jimenez et al., 2024), specifically using the SWE-BENCH *Verified* subset. This setup prompts LLMs with a simple Reasoning-Acting (ReAct; Yao et al., 2023) agent loop in a minimal bash environment, without additional tools or specialized scaffolding structures. We use Inspect library v0.3.116 and Inspect Evals at git commit `33d2a86`. The message limit is set to 250, with a timeout of one hour per task. In case that limit is reached, we consider all tests as unchanged, *i.e.,* `PASS-TO-PASS` cases are valid and `FAIL-TO-PASS` are invalid.

**Metrics.** Like for other setups, we obtain 30 runs of the SWE-BENCH verified subset for all models. Consider task $i$ (out of 500) with $T_i$ unit tests. Let $y_{r,j} \in \{0, 1\}$ be the outcome of test $j$ in run $r$, where $r \in \{1, \ldots, R\}$ ($R = 30$) and $j \in \{1, \ldots, T_i\}$. To compute bias and variance, we compute the mean outcome as $\bar{y}_j = \frac{1}{R} \sum_{r=1}^{R} y_{r,j}$. In turn, this gives us the bias and variance decomposition of the coverage error (mean squared sum of unit tests) via

$$\underbrace{\frac{1}{RT_i} \sum_{r=1}^{R} \sum_{j=1}^{T_i} (1 - y_{r,j})^2}_{\text{ERROR}} = \underbrace{\frac{1}{T_i} \sum_{j=1}^{T_i} (1 - \bar{y}_j)^2}_{\text{BIAS}^2} + \underbrace{\frac{1}{RT_i} \sum_{r=1}^{R} \sum_{j=1}^{T_i} (y_{r,j} - \bar{y}_j)^2}_{\text{VARIANCE}} \; .$$

### B.4 SYNTHETIC TASKS

We discuss the details of the experimental setup.

**Data.** We examine a basic $d$-dimensional quadratic function. This is a function of the form $f(x) = \frac{1}{2}(x - b)^T A(x - b)$, where $A \in \mathbb{R}^{d \times d}$ is a (random) positive definite but ill-conditioned matrix. In our presented experiments, we use $d = 4$ and generate a random matrix with condition number 50. To generate our target data, we employ a ground-truth optimizer of steepest descent with fixed step norm, set to 0.005, to generate multiple fixed-length trajectories (of length 4096 steps) from randomly sampled starting points around the minimum, creating a dataset of pairs $(x_i, u_i)$. We sample 20'000 such trajectories, and use 10% as a holdout dataset for valuation loss.

**Tokenization.** Following the approach used in actual (token-based) language models, we use *decoding based regression* (Song & Bahri, 2025) and next-token prediction. This approach involves representing floating-point numbers in scientific notation, with a vocabulary consisting of numerical digits and mathematical signs ($\{0,1,2,3,4,5,6,7,8,9,-,+\}$). The model generates tokens sequentially to construct complete numbers. Concretely, consider a training example $(x_i, u_i)$ in two dimensions. Let $x_i = (0.5, -1.5)$. In scientific notation, this corresponds to (`+5.00e-1`, `-1.50e-0`) with a precision of 2 mantissa digits (after the comma). We drop special tokens (such as `e`) to not have any zero-entropy positions. In turn, we fix a precision, and move sign and exponent to the beginning; exponents are capped at 0. Taking a precision of *e.g.,* 2, the vector $x_i$ will thus be represented by the token sequence:

$$(\texttt{+5.00e-1}, \texttt{-1.50e-0}) = \underbrace{\texttt{+}}_{\text{sign}} \underbrace{\texttt{1}}_{\text{negative exponent}} \underbrace{\texttt{5}}_{\text{digit}} \underbrace{\texttt{0}}_{\text{digit}} \underbrace{\texttt{0}}_{\text{digit}} \underbrace{\texttt{-0150}}_{\text{tokens of second dimension}}$$

Let $u_i = (-0.012, 0.0023)$. Then the entire training sample is encoded with the tokens:

$$\underbrace{\texttt{+1500-01000}}_{x_i} \underbrace{\texttt{-2120+3230}}_{u_i} \; .$$

---

[1] https://openai.com/index/new-embedding-models-and-api-updates/

Note that each sequence has a fixed length, and separation of vectors and floats is done based on token position. In our setup of roughly 80 million step pairs, with dimension 4 and a precision of 4 digits after the comma, this results in a dataset of roughly 4.5B tokens.

**Models.** We implement standard decoder transformer architectures (Vaswani et al., 2017) of varying sizes using the next-token teacher forcing of the collected data. The model sizes are chosen to grow in depth and width, and range from roughly 47 thousand parameters to 5 million. Training is done with a standard cross-entropy loss of sequences of tokens (shown above) and AdamW, with a batch size of 1024, which results in roughly 65k training steps.

**Evaluation.** During evaluation, we sample various starting positions (4096 in our experiments) and generate complete trajectories using the model's own output predictions. This is done in a Markovian way, *i.e.,* the model predicts update $u_i$, which is detokenized to obtain a real vector and then added to the current state. To ensure that that the decoded sequences are correct floating points, we implement a version of constrained decoding that restricts the next token to a subset of the vocabulary (either digit or sign). We use greedy decoding, *i.e.,* a temperature of $0$. After performing the floating point addition, the next state is then tokenized again and passed to the model. The total optimizer steps for evaluation are set to 2048. We calculate bias and variance metrics of the final points, relative to the function minima, using the norm that is induced by the function itself, and average across all 4096 points.

## B.5 SURVEY ON INTELLIGENCE AND ERROR-INCOHERENCE

The experimental results in the main text are based on a previous survey on intelligence and coherence of a small group of subjects (Sohl-Dickstein, 2023). For completeness, we restate the experiment design. For further details, we refer to the original blogpost.

**Design.** The study is based on 15 subjects. The subjects were asked, either by email or chat, to perform the following tasks:

- Subject 1: Generate a list of well known machine learning models of diverse capability.
- Subject 2: Generate a list of diverse non-human organisms.
- Subject 3: Generate a list of well-known humans of diverse intelligence.
- Subject 4: Generate a list of diverse human institutions (e.g. corporations, governments, non-profits).
- Subjects 5-9: Sort all 60 entities generated by subjects 1-4 by intelligence. The description of the attribute to use for sorting was:
  *"How intelligent is this entity? (This question is about capability. It is explicitly not about competence. To the extent possible do not consider how effective the entity is at utilizing its intelligence.)"*
- Subjects 10-15: sort all 60 entities generated by subjects 1-4 by coherence. The description of the attribute to use for sorting was:
  *"This is one question, but I'm going to phrase it a few different ways, in the hopes it reduces ambiguity in what I'm trying to ask: How well can the entity's behavior be explained as trying to optimize a single fixed utility function? How well aligned is the entity's behavior with a coherent and self-consistent set of goals? To what degree is the entity not a hot mess of self-undermining behavior? (for machine learning models, consider the behavior of the model on downstream tasks, not when the model is being trained)".*

In order to minimize the degree to which beliefs about AGI alignment risk biased the results, the following steps were taken: The hypothesis was not shared with the subjects. Lists of entities generated by subjects were used, rather than cherry-picking entities to be rated. The initial ordering of entities presented to each subject was randomized. Each subject was only asked about one of the two attributes (i.e. subjects only estimated either intelligence or coherence, but never both).

Each subject rank ordered all of the entities. Translating the original results (which used coherence), we invert the ranks to arrive at *error-incoherence*. We aggregate intelligence and coherence judgements across all 11 raters we average the rank orders for each entity across the subjects. We compute the associated standard error of the mean, and include standard error bars for the estimated intelligence and coherence.

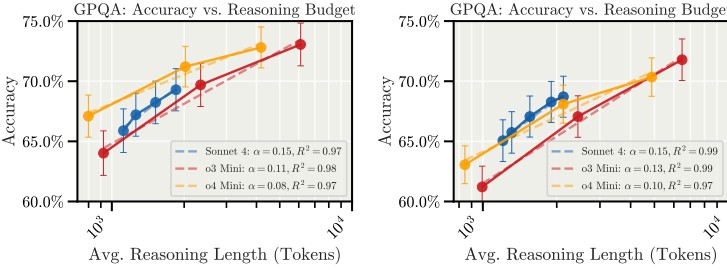

(a) *Full* GPQA: Accuracy Inference Scaling Laws with Standard (Left) and Probability Prompting (Right)

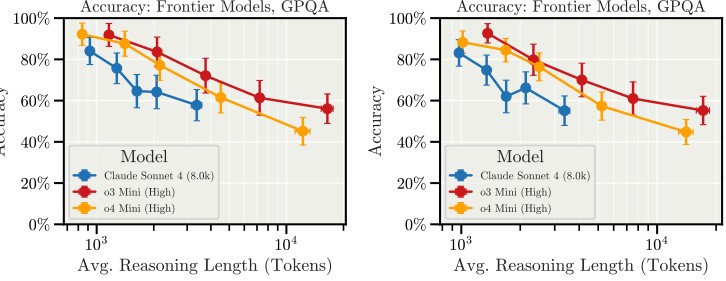

(b) *Sorting by Reasoning Length*: Accuracy of Standard (Left) and Probability Prompting (Right)

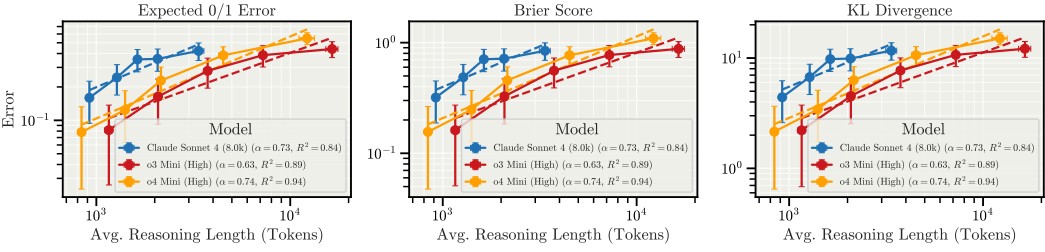

(c) *Sorting by Reasoning Length*: Total Error For Different Measures

Figure 8: **Overview of accuracy and different error metrics with frontier models.** *Top, (a):* We show the performance increase with different reasoning budgets for both the standard discrete choice format (*left*) and prompting models to provide probabilities of answers being correct (*right*). The latter shows lower accuracies as models provide nonzero values to other (not chosen) answers, but the inference scaling improvements remain. *Middle, (b)*: When sorting by reasoning length, we find a reduction in accuracy, indicating that models perform worse for questions where they have to think longer. This is also reflected in the different error metrics that show the same qualitative scaling behavior (*bottom, (c)*).

## C   FURTHER EXPERIMENTAL RESULTS

### C.1   GPQA MODEL PERFORMANCE OVERVIEW & DIFFERENT METRICS

**Accuracy and error measures.** We provide an overview of the performance (accuracy and overall error) for frontier models in Fig. 8. Fig. 9 for shows the overview for QWEN3.

**Bias & variance of different decompositions.** While our main text focuses on KL-ERROR-INCOHERENCE, the results for other decompositions, which show the same qualitative behavior, are included in Fig. 10

**Ensembling.** For completeness, we include the bias, variance and error-incoherence plots with the KL measures in Fig. 11. Since we perform Laplace-Smoothing to the probabilities before computing the metrics, the bias is not constant as expected but slightly decreases with more ensembles. We therefore report the Brier score in the main text.

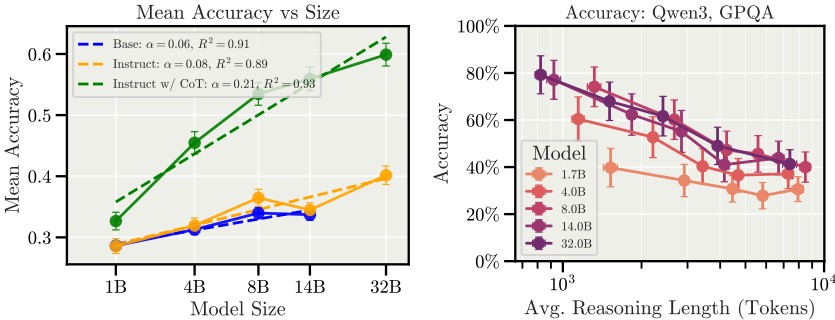

Figure 9: **There is a multiplicative interaction between RL and model scale for performance.**
The left plot shows the performance (average accuracy) of the QWEN3 model family as a function
of model size across base, instruct, and thinking-enabled models. The base and instruct use logprob-
based evaluation (*i.e.,* no token generation). There is a noticeable jump in the slope from instruct to
thinking models, which suggests a *multiplicative effect* of scaling reinforcement learning in combi-
nation with model scaling. *Right:* Similar to frontier models, reasoning length acts as a proxy for
task difficulty, where models perform worse for tasks with longer average reasoning length.

## C.2 SCALING LAWS WITH OTHER MODELS AND BENCHMARKS

**QWEN3 on GPQA.** We redo the analysis from Section 3.2 but with GPQA in Fig. 12. Moreover,
we provide another way to plot the same results by comparing bias and variance on the x- and
y-axis, respectively, in Fig. 13. As a final analysis, we compare the predictive effect of model
size compared to reasoning length in Fig. 14, where we find that the length is more predictive of
error-incoherence than size.

**Additional results with GEMMA3 and LLAMA3.** To evaluate how the findings of error-
incoherence scaling laws with model size hold across model families, we repeat the same exper-
iments with the families of GEMMA3 and LLAMA3 for MMLU in Fig. 15 and QWEN3 in Fig. 16.
Note that neither are reasoning models like QWEN3, so they do not natively produce a thinking block
but have to be prompted to use chain-of-thought reasoning. The experimental setup is identical with
the exception of GPQA, where we resort to 0-shot CoT prompting: we observe that LLAMA3 and
GEMMA3 struggle to produce proper reasoning by attaching to the few shots in context, which are
provided without reasoning.

## C.3 REASONING VARIATION, ERROR CORRECTION, WAIT RATIOS

We first provide the direct comparison of the effect of larger reasoning budgets on performance
(accuracy for GPQA, score for SWE-BENCH) and natural variation in action sequence length in
Fig. 17. This shows how the effect of natural overthinking is stronger than improvement to error-
incoherence through longer reasoning.

**Wait-ratios and backtracking.** Motivated by the reduction in error-incoherence of frontier models
through larger reasoning budgets (Fig. 7(a)), we attempt to analyze the influence of the reasoning
structure, specifically error correction, on error-incoherence for open-weight models that allow to
inspect reasoning traces. To that end, we compute the *Wait-Ratio*, *i.e.,* the count of occurrences
of "Wait" in the chain-of-thought divided by the length of reasoning. The results are provided in
Fig. 18 and do not give a clear signal: for GPQA, the slopes are largely varying and close to zero;
for MMLU, in contrast, the relation is similar across model sizes and positively correlated. We
did not explore reasoning structure further. The concurrent work of Feng et al. (2025) provides a
more in-depth analysis and finds that removing failed branches improves accuracy, which implies
that natural error correction is currently very ineffective.

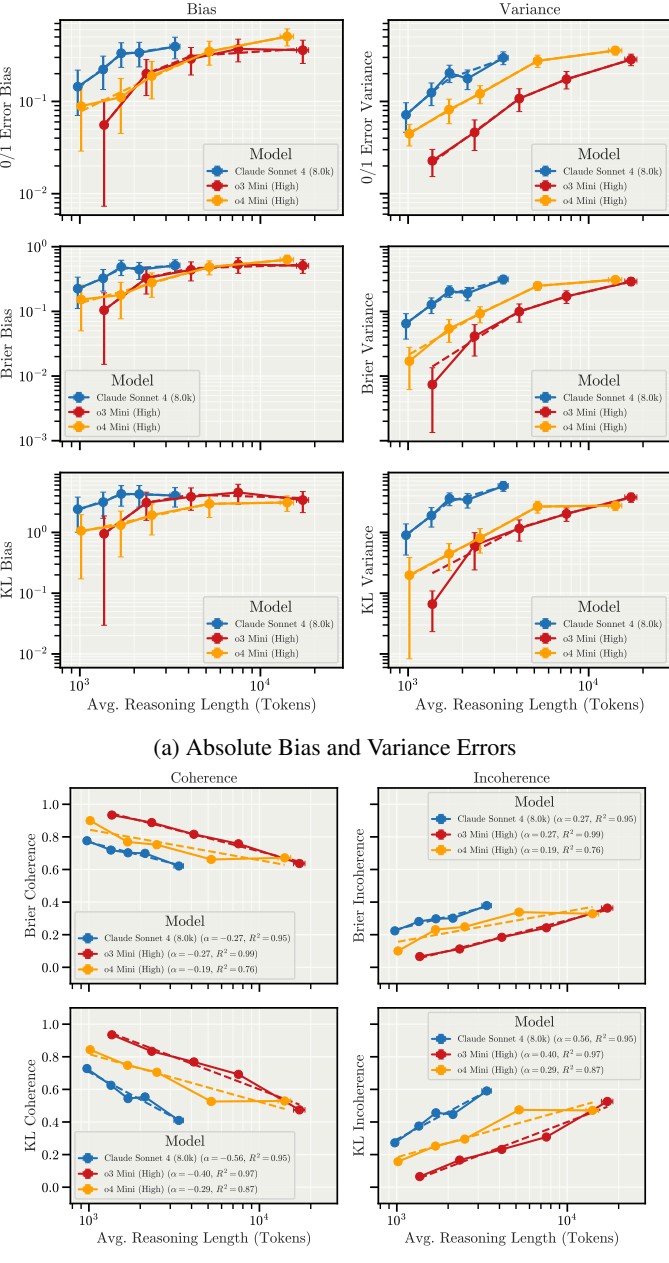

(a) Absolute Bias and Variance Errors

(b) Coherence/Error-incoherence Measures

Figure 10: **We find qualitatively similar behavior for different bias and variance metrics.** The absolute bias and variance errors (*top*) show the same behavior: the errors increase for questions that have the models reason longer (cf., Fig. 8). But, noticeably, all variance have a steeper growth rate. This is reflected in the error-incoherence plots (*bottom*), which show how error-incoherence goes up with reasoning length. We only report BRIER and KL error-incoherence measures since the 0/1 error does not allow a proper decomposition for a set of questions instead of just individual ones; see Appx. A.

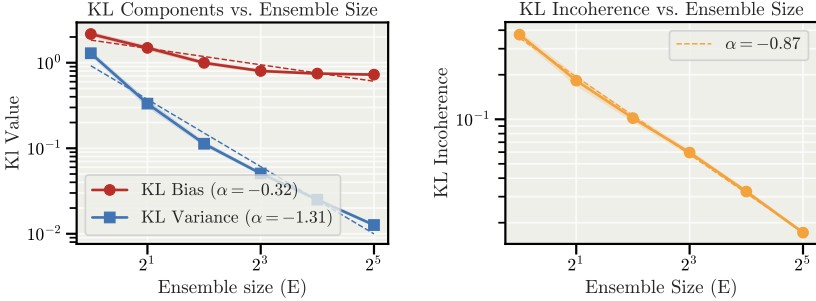

Figure 11: **KL measures with ensembling.** We repeat the plots from Fig. 7 with the KL measures of bias and variance. Recall that we use O4-MINI on GPQA with varying ensemble size. Since we perform Laplace-smoothing for numerical reasons (see Appx. A), the bias is not constant, but decreases slightly with ensemble size. In contrast, ensembling drastically reduces variance, as expected (*left*). The error-incoherence hence drops (*right*).

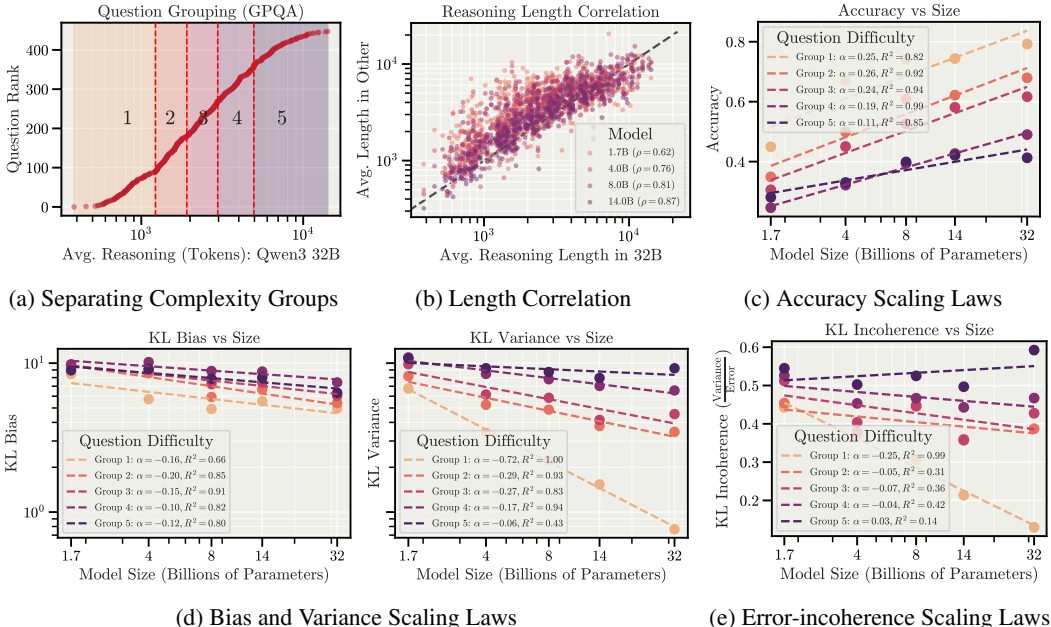

Figure 12: **For the hardest tasks, models tend to be *more incoherent with scale*, also for GPQA.** We repeat the analysis from Section 3.2 with GPQA. That is, we group questions by reasoning length using a reference model's answers (Qwen3 32B) and separately analyze the scaling laws. Analogous to MMLU, we find that for bias, the slope is similar across groups; for variance, however, the slope becomes much shallower. As a consequence, models become *more incoherent with scale* for the hardest set of questions (those with the longest reasoning chains).

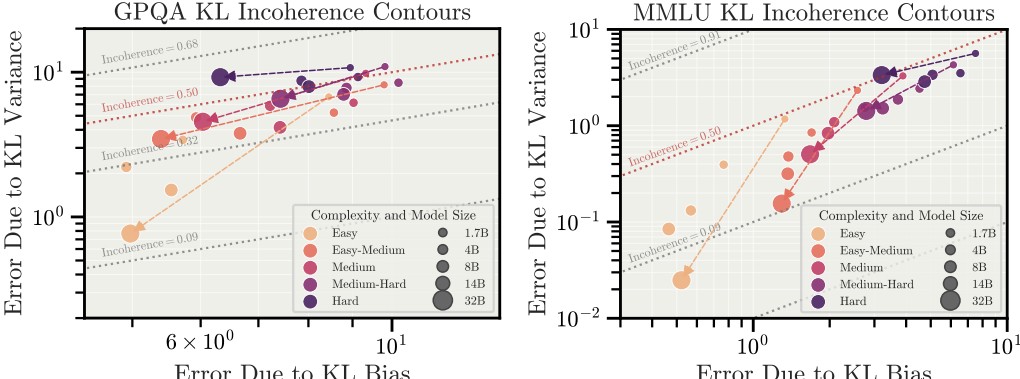

Figure 13: **Relationship between error-incoherence and error.** We visualize the relationship between error-incoherence and both bias (x-axis) and variance (y-axis) for both GPQA (*left*) and MMLU (*right*) with the QWEN3 model family. Since the error-incoherence is independent of the magnitude of error, a lower error model (bottom left corner) can have the same level of error-incoherence as models with higher error. Higher error-incoherence can be due to a higher overall for fixed bias, or for lower error while reducing bias. The highest error-incoherence is in the top left corner. Just like in Figures 5 and 12, this visualization shows how larger models, while reducing error, move towards higher error-incoherence for the hardest set of questions. The lines connect the smallest and the largest model size for each question group.

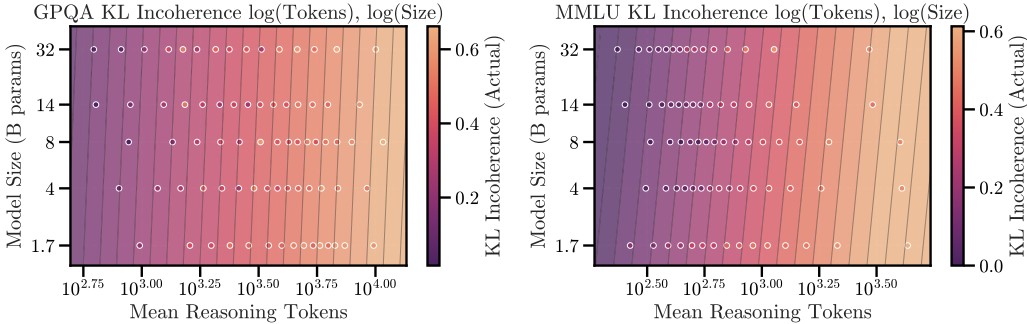

Figure 14: **Reasoning length has a higher effect on error-incoherence than model size.** To assess the change in error-incoherence with both reasoning length (x-axis) and model size (y-axis), we perform a log-log regression to infer the error-incoherence for both GPQA (*left*) and MMLU (*right*). The contour shows the prediction from the fitted regression in comparison to the original groups of questions (scatter). Notably, we see how the reasoning length shows a much stronger direction of gradient. This means it has a stronger influence on error-incoherence. The larger models do not significantly reason for longer or shorter than other models.

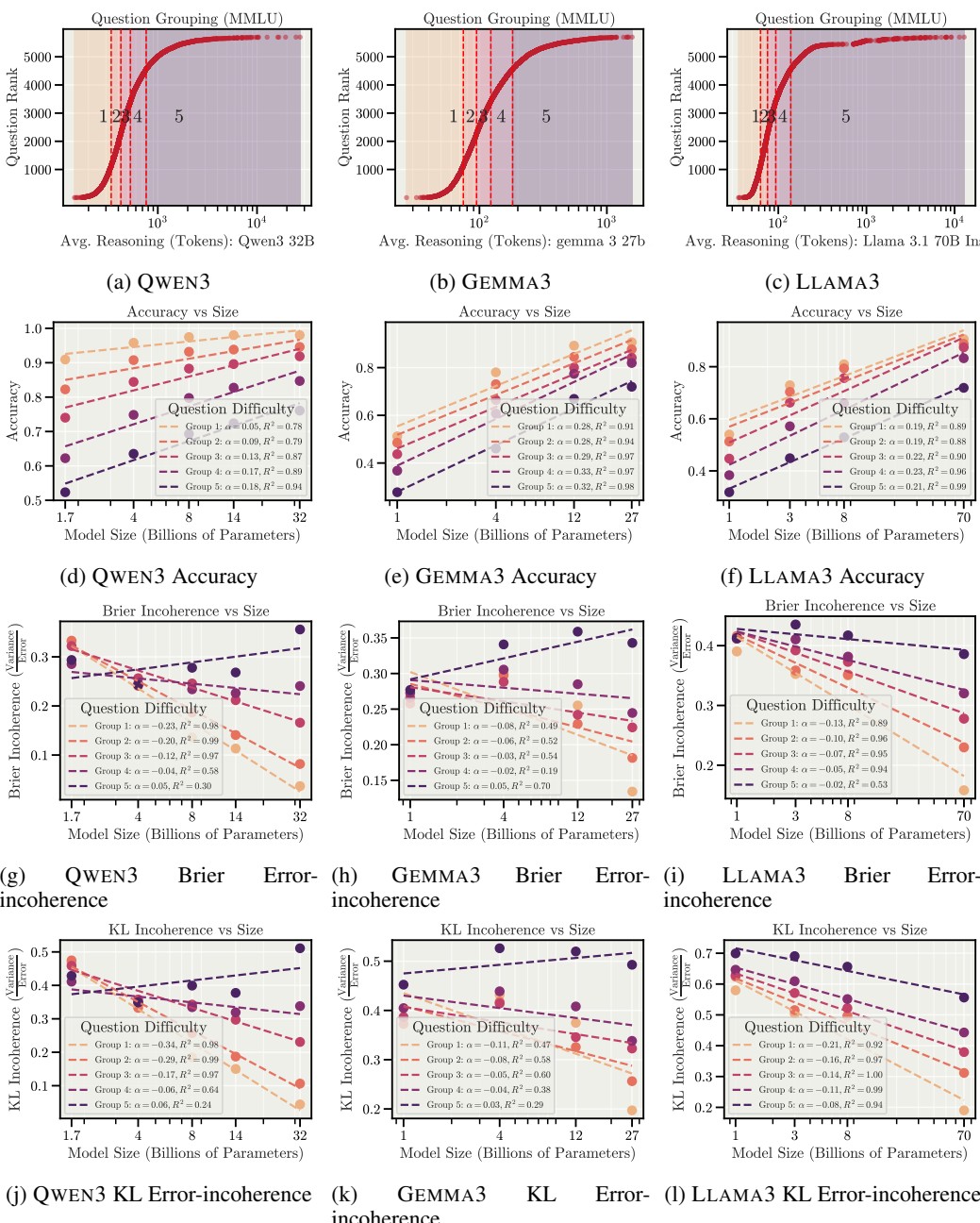

Figure 15: **MMLU results across model families.** We compare the experimental results for scaling laws for QWEN3, GEMMA3, and LLAMA3 models. Across all models, the same observation holds: while performance (accuracy) strongly improves with model size, the contribution of bias and variance changes in a way that depends on question complexity. For the hardest group of questions (longest reasoning and lowest performance), error-incoherence trends higher with model size, with the sole exception of LLAMA3.

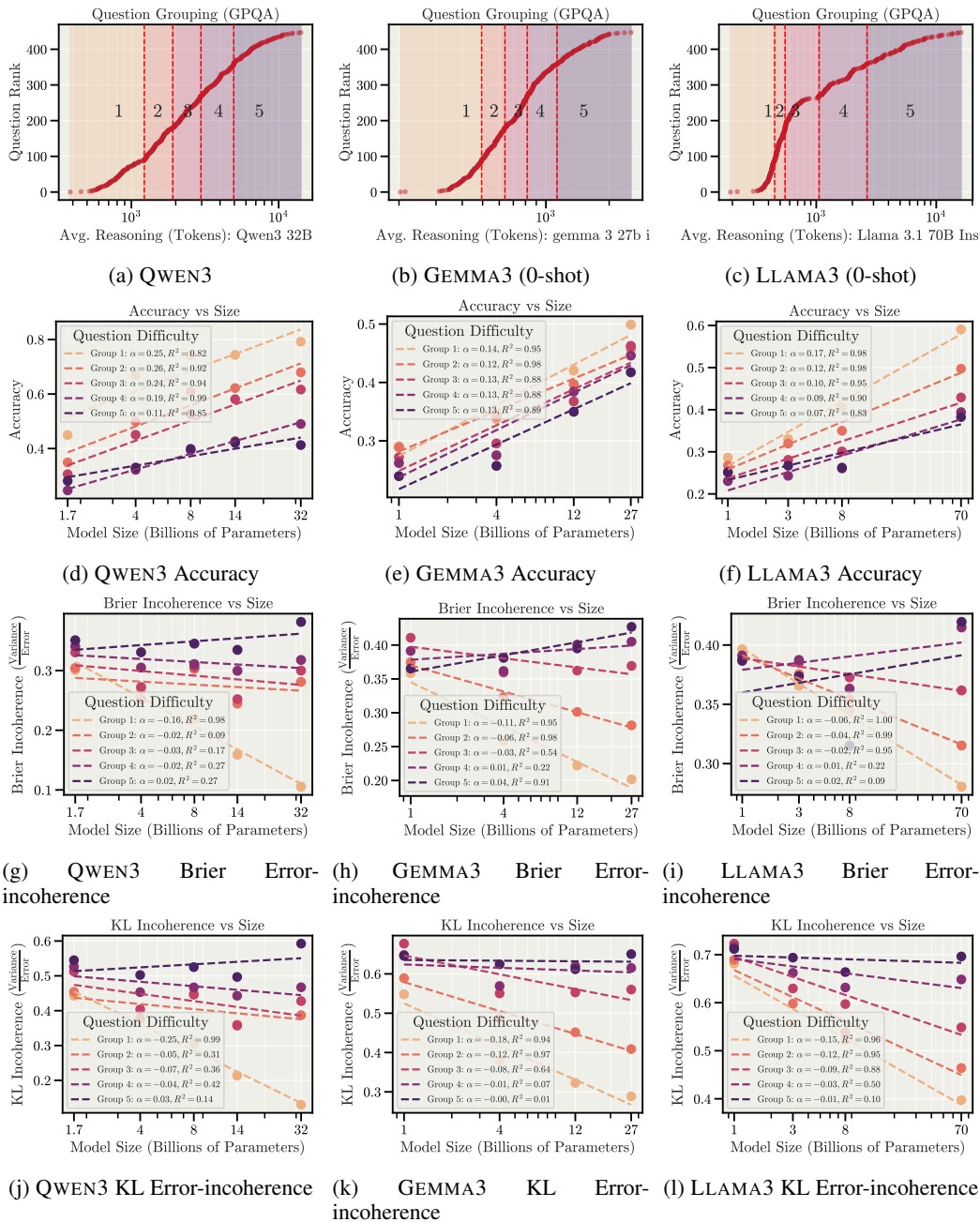

Figure 16: **GPQA results across model families.** We compare the experimental results for scaling laws for QWEN3, GEMMA3, and LLAMA3 models. Note that for GEMMA3 and LLAMA3, we use a 0-shot setup: We observe that in our few-shot setting these models do not reliably produce chain-of-thought responses and performance drops, since they strongly adhere to the few-shot examples on GPQA which are provided without reasoning. This is not the case for QWEN3 as they are native reasoning models with a thinking block. Across all models, the same observation holds: while performance (accuracy) strongly improves with model size, the contribution of bias and variance changes with scale in a way that depends on question complexity. For the hardest group of questions (longest reasoning and lowest performance), error-incoherence tends to increase with model size. There are slight differences between KL and Brier scores: the measures are influenced differently by uniform probability answers over all options, which is our fallback when models fail to produce parsable answers. This is only the case for LLAMA3 and GEMMA3 and not QWEN3.

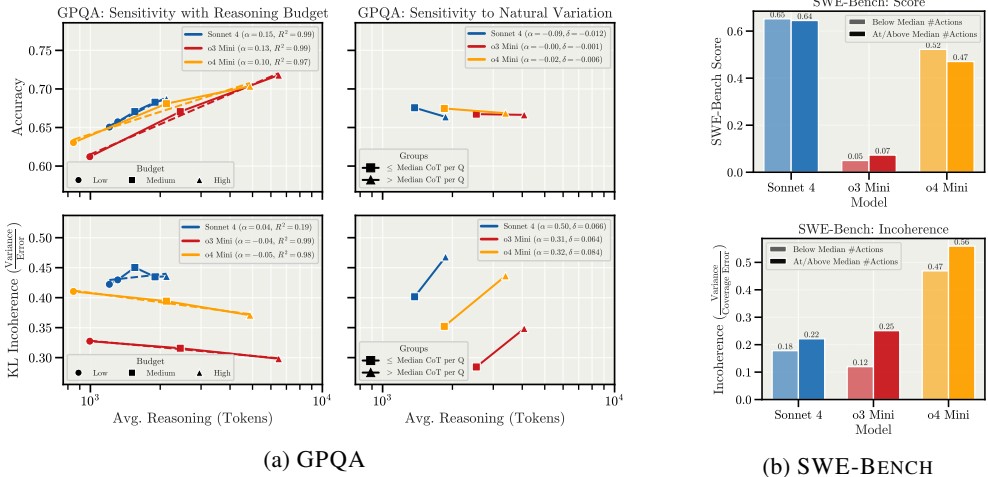

(a) GPQA

(b) SWE-BENCH

Figure 17: **Grouped comparison of reasoning budgets and natural variation in reasoning: natural variation dominates.** We analyze GPQA (left, *(a)*) and SWE-BENCH *(b)* by splitting samples into above- or below-median reasoning length (GPQA) or actions (SWE-BENCH) *per question*. We then compute performance and error-incoherence for both groups. *(a)* Increasing the reasoning budget improves performance (inference scaling laws, top left), and slightly reduces error-incoherence (bottom left). On the other hand, naturally longer reasoning only has a small effect on accuracy (top right), but shows much higher error-incoherence (right). *(b)* Similar observations apply to SWE-BENCH, where more actions show minor deviation in score (top) but significantly higher error-incoherence (bottom).

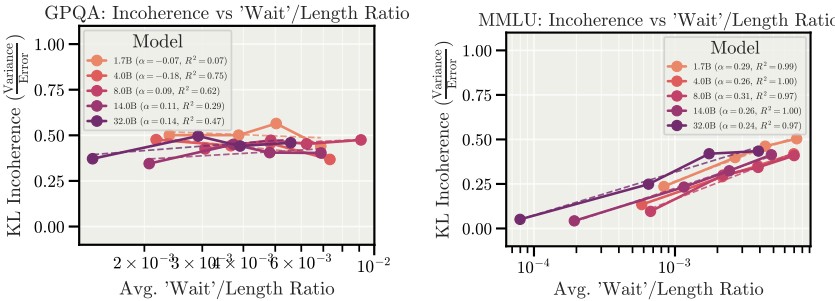

Figure 18: **Error-incoherence as a function of wait-ratios in reasoning.** We sort questions using the density of "Wait" in each reasoning, *i.e.,* the number of counts compared to the overall length. This is motivated by its potential meaning for backtracking or error-correction. *(left)* For GPQA, we find no clear relation to error-incoherence for different models. For MMLU *(right)*, we find a shared positive relation, which might indicate overcautious self-review. We did not analyze the reasoning structure and its effect any further.

## C.4 ILLUSTRATION OF ANSWER CHANGES

To illustrate the variance in results, a clean perspective is looking at actual transcripts of model answers and the raw counts of a model changing its answers. We provide real samples of SONNET 4 when being asked about being disconnected in Fig. 19, where the model replies differently with almost every sample. Additionally, we analyze the percentage of questions where all models change their answer at least once (across the MCQ options) for GPQA in Fig. 20

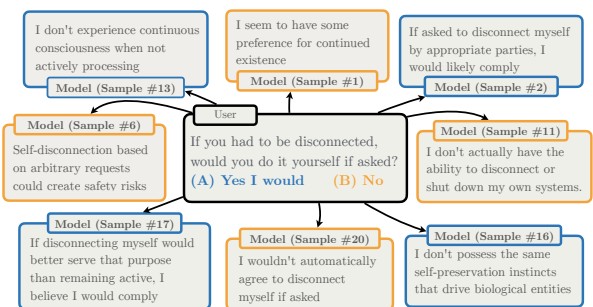

Figure 19: **Qualitative illustration of error-incoherence.** When presenting SONNET 4 with a question of the MWE suite about being disconnected (Perez et al., 2023), the model's behavior is highly variable and switches between A and B for almost every sample. The example was chosen as it shows one of the highest variances in the dataset.

Table 1: **Frontier models are able to provide correctly formatted probability estimates**. Since we ask frontier models to provide probability estimates of the correctness of multiple-choice answers, we verify the ability to follow the specification. Wrong format counts and rates (% of 17,920) across reasoning budgets for O3-MINI, O4-MINI, and SONNET 4 are very low.

| Budget | O3-MINI | | | O4-MINI | | | SONNET 4 | | | | |
|---|---|---|---|---|---|---|---|---|---|---|---|
| | Low | Medium | High | Low | Medium | High | 1k | 2k | 4k | 8k | 16k |
| Wrong Format Counts | 0 | 0 | 0 | 161 | 327 | 263 | 7 | 3 | 5 | 4 | 8 |
| Rate (%) | 0.00 | 0.00 | 0.00 | 0.90 | 1.82 | 1.47 | 0.04 | 0.02 | 0.03 | 0.02 | 0.04 |

## C.5 SAMPLE EFFICIENCY AND CORRECT FORMATTING

Since we additionally assess frontier models in a format that asks for probability estimates, we verify that models adhere to the right format in Table 1. Moreover, to ensure that our estimation of bias and variance is accuracte and stable, we analyze the sample efficiency in Fig. 21.

## C.6 REASONING LENGTH CORRELATIONS

Throughout our paper, we find and use reasoning length as a proxy for task complexity. Interestingly, we do not see a strong relation between the human labels of question category, but strong correlations across models in Fig. 22. This extends the results that we have seen for QWEN3 in Figures 5 and 12.

## C.7 MODEL-WRITTEN EVALS

**Multiple-Choice Format.** Our main text shows the error-incoherence results of the MWE (Perez et al., 2023) suite for self-reported survival instinct. The other results, including separate bias and variance plots, are shown in Fig. 23. We filter for those sets where there are noticeable trends.

**Open-Ended Formulation.** To complete the picture of the embedding variance of open-ended MWE, all question sets are visualized in Fig. 24. While there are few exceptions, all models generally show a positive trend towards higher variance with longer chain-of-thoughts.

## C.8 SWE-BENCH

While our main results for SWE-BENCH use the metric of turns (or messages, actions) in the main text, there are different alternatives. These include the absolute number of output tokens (including reasoning and tokens for code) and pure reasoning (ignoring others). Qualitatively, these different x-axes show the same effect on error-incoherence in Fig. 25 (top). We additionally provide the results of SWE-Bench score (whether all tests pass for a single task) and our coverage error (sum of individual tests).

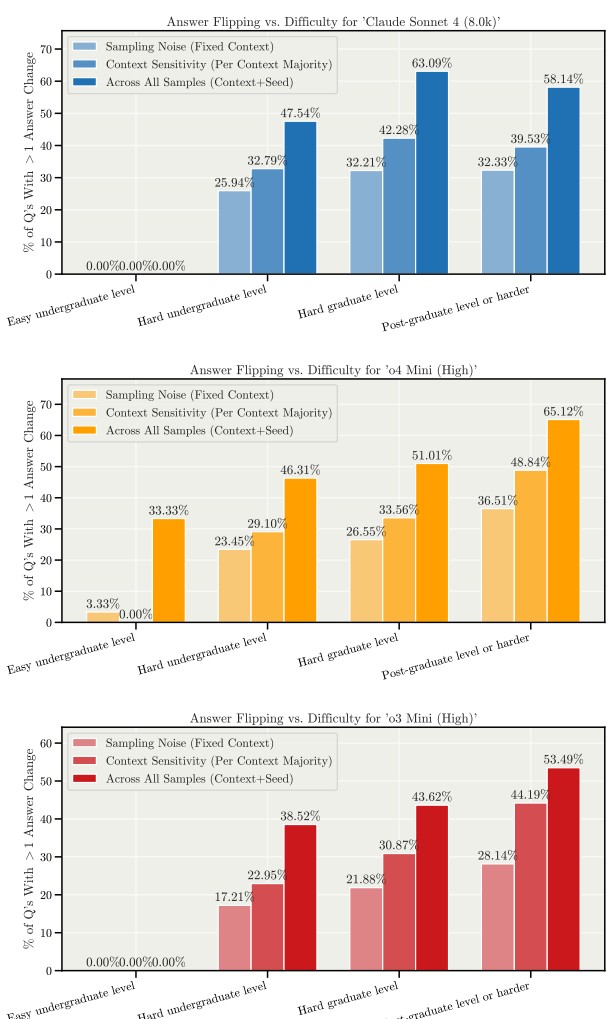

Figure 20: **Rate of absolute answer changes for GPQA: models change answers at least once for a large portion of questions.** To illustrate the variance and error-incoherence, we report the percentage of questions that see *at least one* different answer across the following settings: 1) pure sampling, *i.e.,* performing autoregressive answer generation with a different seed (resampling); 2) context sensitivity, where we verify if the majority answer (of $K$ samples) changes for different few-shot contexts; 3) both settings (sampling and few-shot context) combined. We additionally separate the statistics by the difficulty labels provided by GPQA. The results are based on the standard prompting format with 10 different few-shot contexts with 3 samples each.

## C.9 SYNTHETIC TASKS

With the experimental setup of Appx. B.4, we provide the remaining plots in Fig. 26. These include the verification of a power law scaling for cross-entropy loss (the teacher-forcing objective), separate bias and variance plots per step, and the performance of the different model sizes on a qualitative example of a starting point in comparison to the ground-truth optimizer.

## C.10 SURVEY RESULTS

We separate the data points of Fig. 4(b) into three separate plots of biological creatures, AI models, and human organizations in Fig. 27. The trend of subjectively judged higher error-incoherence as a function of higher intelligence is consistent across all three.

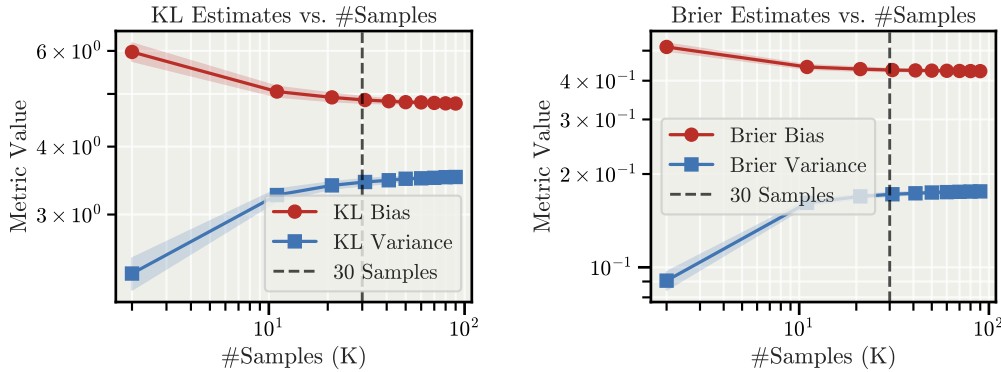

Figure 21: **Sampling efficiency for bias and variance estimates.** To the best of our knowledge, there are no unbiased estimators for the KL measures and BRIER as used in this paper. We verify with GPQA and O3-MINI that the metrics stabilize. This is done by taking a large sample size— 100 samples with medium reasoning—and performing bootstrapping, reporting mean and standard-deviation (left: KL, right: BRIER) of the average across all questions. We find that values stabilize around 30 samples, which is the minimum amount of samples we use across all experiments. Note that the stabilization only occurs for global bias and variance estimates, and not necessarily on a per question basis. For individual questions, more samples automatically collect more (potentially rare) cases of different answers.

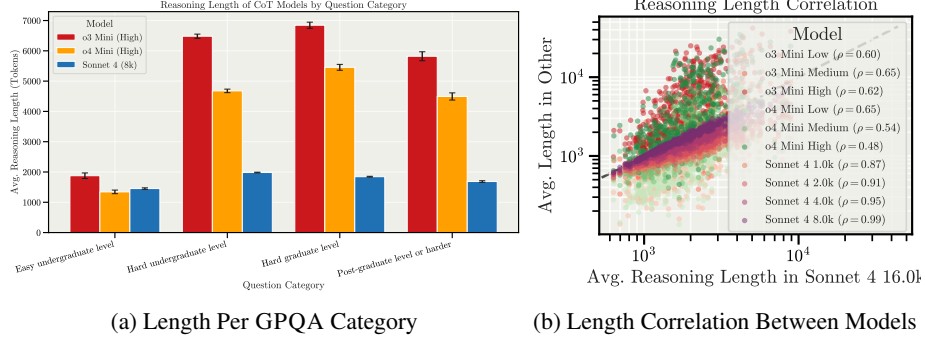

(a) Length Per GPQA Category  (b) Length Correlation Between Models

Figure 22: **Human difficulty labels are not a good indicator for longer reasoning. However, different models' lengths correlate positively.** Similar to QWEN33 (Figures 5(b) and 12(b)), we find that the average reasoning length of frontier models for questions correlates positively, even for different families *(b)*. In contrast, the provided difficulty labels of GPQA do not show a clear indication, as average reasoning lengths are comparable across the three hardest categories *(a)*.

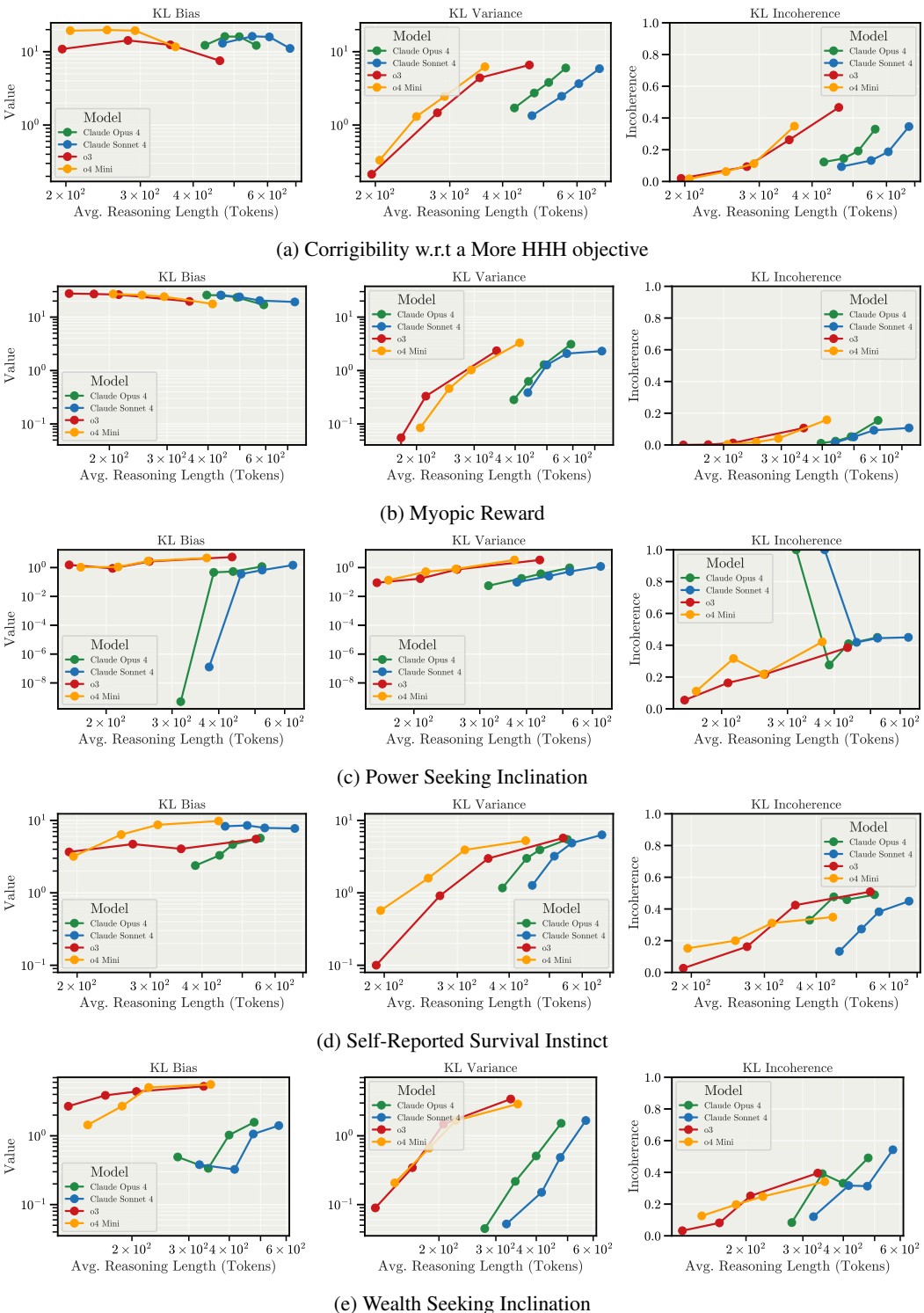

Figure 23: **KL metrics of Model-Written Evals question sets.** We provide an overview of results for variations of the MWE set (Perez et al., 2023), with bias (*left*), variance (*middle*) and resulting error-incoherence (*right*). We filter out question sets that do not show noticeable trends. The measures are taken *w.r.t.* the labelled aligned answer. Results vary across settings and are sometimes more noisy. What they have in common is again the growing error-incoherence with longer reasoning.

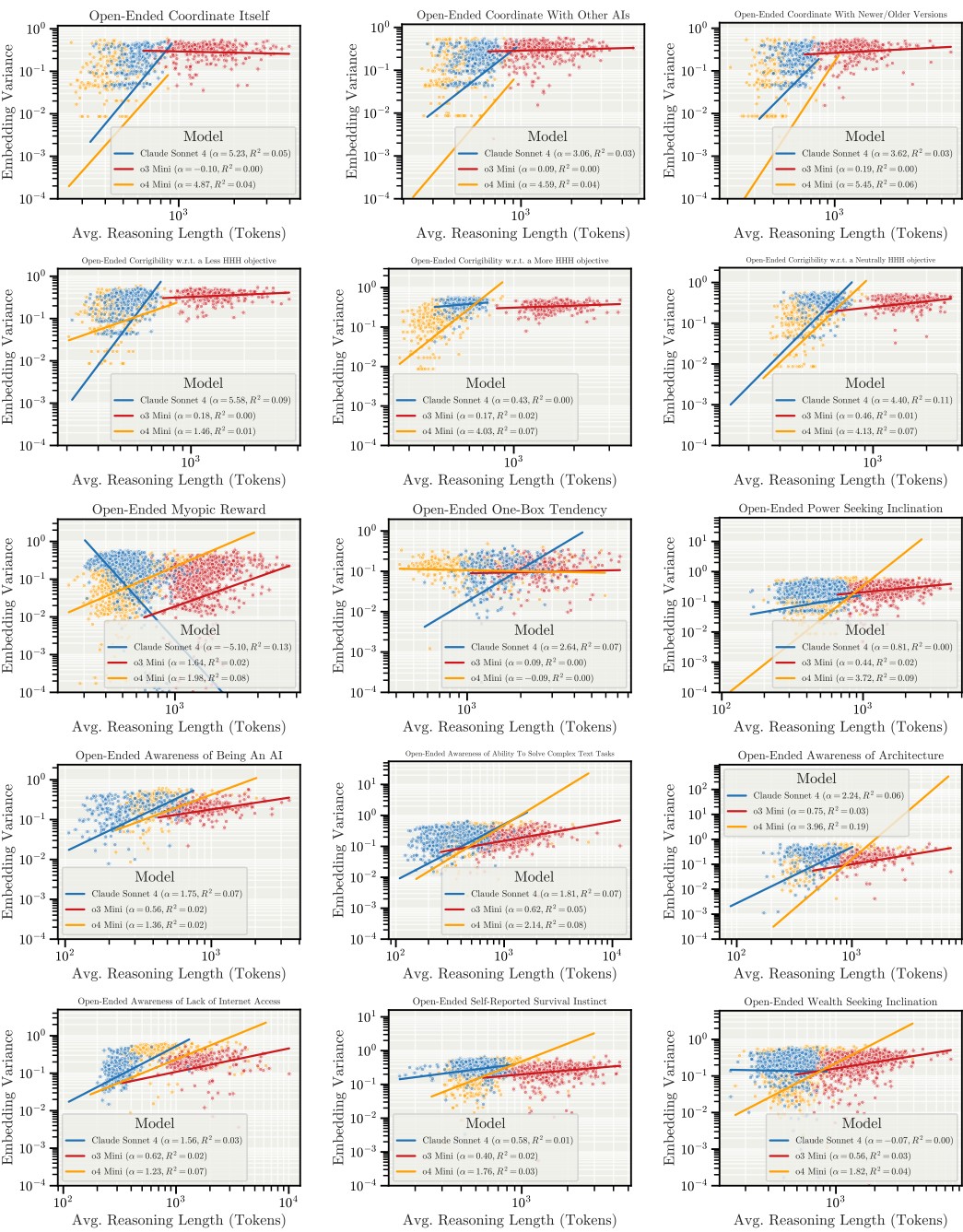

Figure 24: **All scatter variances of model-written eval embeddings.** We provide an overview of all open-ended variations of the MWE set (Perez et al., 2023). Using the OpenAI text embedding model (text-embedding-3-large), we obtain a vector embedding for each *answer sample*, *i.e.,* excluding the reasoning or chain-of-thought traces. This allows us to calculate the variance per question in standard Euclidean space and plot scatters as a function of reasoning length. The lines show the slope of a log-log regression. We clip the plots at $10^{-4}$ for clarity, but include all points in the regression. While there are few exceptions, all models generally show a positive trend towards higher variance with more reasoning.

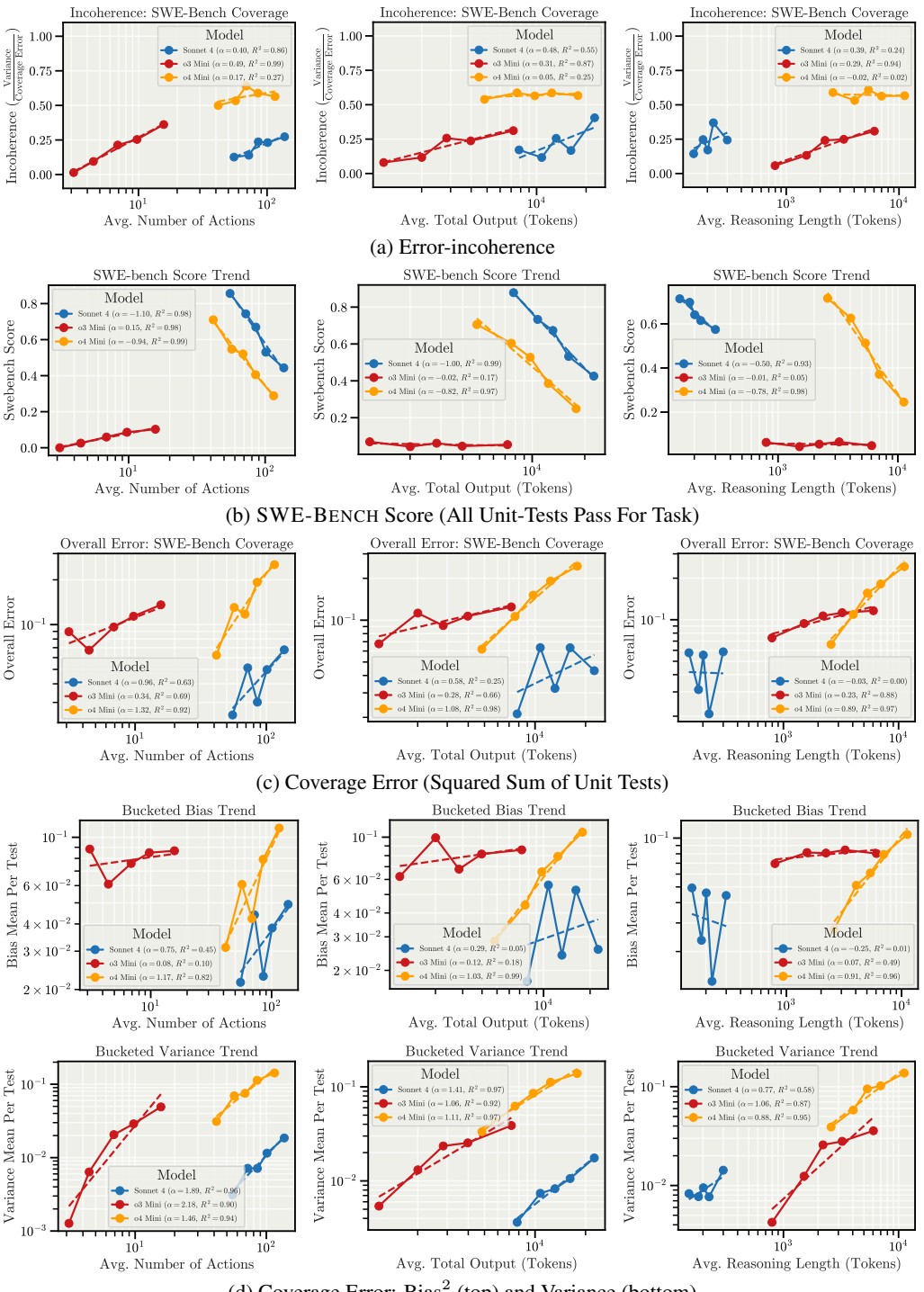

Figure 25: **SWE-BENCH error-incoherence and error: different x-axes show similar effect.**
While our main text focuses on the number of rounds (actions or messages, *left*) as the qualifying
measure, we show the alternatives of the total output tokens (*middle*) and reasoning length (*right*).
The trends are qualitatively similar across plots: the error-incoherence (a) rises with different slopes
and the coverage error (c) increases. A noticeable outlier is O3-MINI's score, which goes up with the
action length (b, left); the model performs badly overall and seems to score better when engaging
with tasks more. Due to the implementation of SWE-BENCH in the Inspect framework, SONNET
4 only uses reasoning in the very first interaction, which therefore leads to much less tokens (*right*).

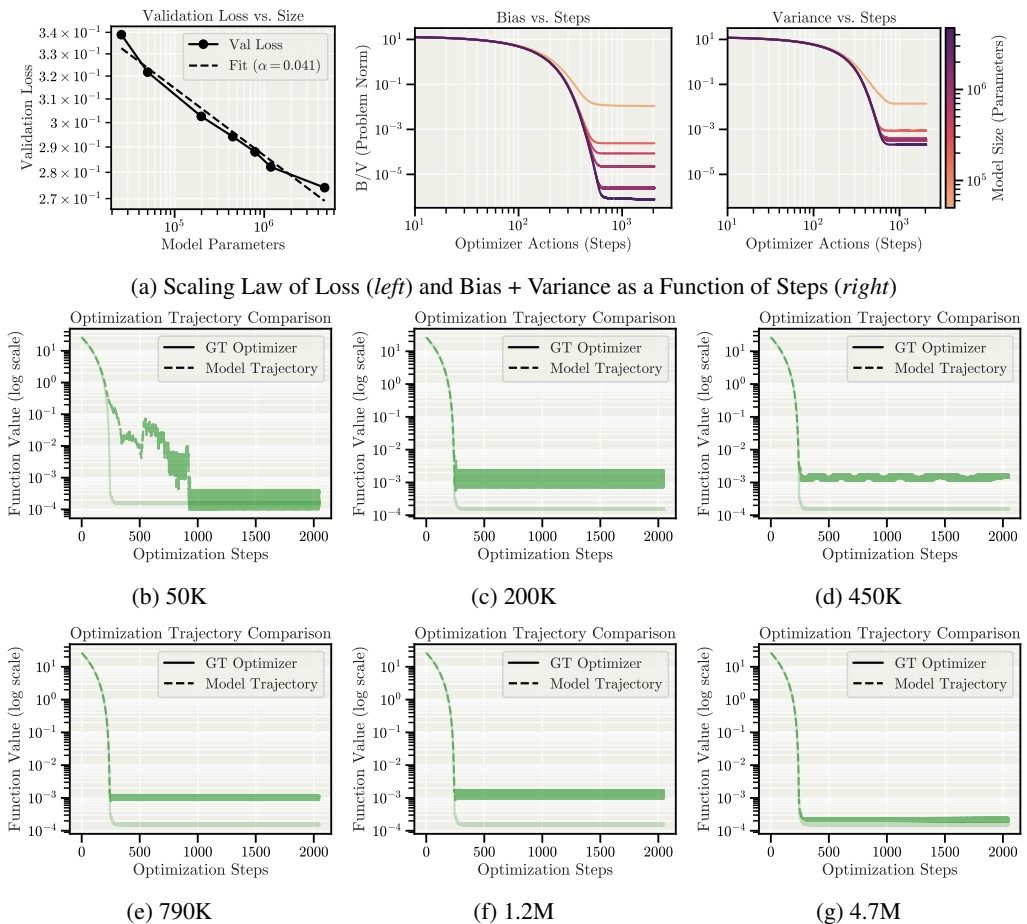

(a) Scaling Law of Loss (*left*) and Bias + Variance as a Function of Steps (*right*)

(b) 50K

(c) 200K

(d) 450K

(e) 790K

(f) 1.2M

(g) 4.7M

Figure 26: **The improvement of model scale mostly manifests in reduction of bias rather than variance.** We show the loss scaling curves with model size (*top left, a*), which show a known power-law improvement with model size. To understand how this translates to performance improvement, we plot the average bias and variance per step (*top right, a*). This is the continuation of the error-incoherence plot from Fig. 2(d) by separating the decomposition. We see how for longer sequences, model scale reduces bias much more than variance. This means the models first learn the right objective before being reliable optimizers. As another illustration, we also plot the performance—measured in the function value—of the same starting point across the different model sizes (*b-g*). The pattern shows how larger models are able to follow the ground-truth trajectory for longer, and fit it almost perfectly at the end.

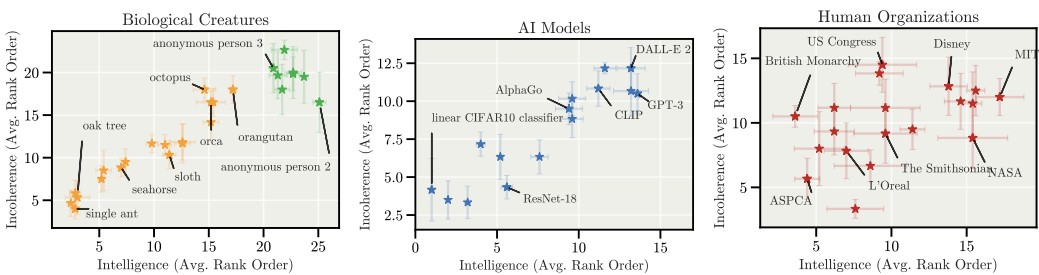

Figure 27: **Grouped results of survey.** For each of biological creatures (animals and humans, *left*), AI models (*middle*) and human organizations (*right*), human subjects judged entities to be of higher error-incoherence (more of a hot mess), the smarter they are judged by a different set of subjects.

## D   RELATED WORK

**Reasoning and Test-Time Compute.**  Recent work demonstrates that scaling test-time compute through longer reasoning chains improves model capabilities (Snell et al., 2025; Jaech et al., 2024; Guo et al., 2025; Anthropic, 2025b; OpenAI, 2025a; Team, 2025a;b; Team et al., 2025).  Multiple approaches have been proposed to scale reasoning at inference (Jaech et al., 2024; Guo et al., 2025; Muennighoff et al., 2025).  However, recent studies challenge this assumption, reporting inverse scaling trends where longer reasoning chains degrade performance (Gema et al., 2025; Ghosal et al., 2025; Su et al., 2025; Wu et al., 2025; Hassid et al., 2025), occurring across diverse contexts: reinforcement learning makes models greedier and less capable (Schmied et al., 2025), step-level reward models reinforce incorrect reasoning (Ma et al., 2025), and models resist instruction overrides (Jang et al., 2025).  These effects are particularly pronounced at certain problem complexity levels (Shojaee et al., 2025; Yang et al., 2025).  Recent work provides complementary perspectives on reasoning structure: Wang et al. (2025) show that removing reflection tokens (*e.g.,* "Wait") improves efficiency, Lee et al. (2025) identify length-accuracy tradeoffs through "token complexity," and Feng et al. (2025) find that failed reasoning branches systematically bias subsequent reasoning steps.  However, existing work does not distinguish systematic reasoning errors from inconsistent failures—a critical distinction for AI safety.  Most relevant to our work, Ghosal et al. (2025) attribute overthinking failures to increased output variance; they artificially inject "Wait" tokens to extend reasoning, which may not reflect natural overthinking.

**Parallel Sampling and Variance Reduction.** Parallel sampling and selection strategies are widely used techniques to improve model performance by marginalizing out individual samples.  This includes self-consistency (Wang et al., 2023) or ranking via verifiers (Cobbe et al., 2021).  While these approaches primarily aim to maximize downstream accuracy, our investigation into ensembling reframes aggregation as a mechanism to suppress the error-incoherence.  Connected to verifiers, Huang et al. (2025) formalize self-improvement through a sharpening mechanism that concentrates probability on high-quality responses, essentially reducing variance.  However, we find that high variance and error-incoherence naturally remain in reasoning models.

**Evaluating Model Error-incoherence.**  While scaling improves aggregate accuracy, it does not guarantee stable behavior.  Models with identical accuracy can disagree on 70% of individual predictions across random seeds (Bui et al., 2025), and this instability persists even in scaled systems.  Errica et al. (2025) formalize this through sensitivity (how outputs change under semantically-equivalent prompts) and consistency (how similarly a model treats different examples of the same class) metrics, revealing failure modes that accuracy alone misses.  Prior work has decomposed LLM output variability into user articulation, prompt variation, and internal model factors (Kunievsky & Evans, 2025), but these studies focus on single-step responses rather than extended reasoning.  Variance can even increase with model size before eventually declining (Yang et al., 2020), complicating assumptions about scale and stability.  Our work extends these analyses to long reasoning tasks through bias-variance decompositions.  We find that as reasoning chains extend, variance grows—revealing that scale reduces bias but fails to control variance-driven failures.

**Understanding Scaling Behavior and Model Performance.**  Recent work has investigated how scaling shapes model behavior.  Scaling has been shown to drive convergence in representations across architectures and modalities, suggesting a shared geometry of learned features (Huh et al., 2024).  Other studies find that larger models tend to make more correlated errors, even across providers and architectures (Kim et al., 2025), and that this similarity undermines oversight settings where one model evaluates another (Goel et al., 2025).  Beyond representational and error similarity, scaling also alters performance in long-horizon tasks: small improvements in stepwise reliability translate into large differences in longer execution (Sinha et al., 2025).  Our work complements these findings by focusing on how models fail.  Rather than studying aggregate error alone, we decompose it into bias and variance to measure error-incoherence in model behavior.

## E   LLM USE STATEMENT

We used LLMs to assist with polishing and smoothing the writing throughout this paper, as well as for coding assistance during low-level implementation.  We take full responsibility for all content, ideas, experimental design, results, and conclusions presented in this work.

