# OpenReview forum: "The Hot Mess of AI: How Does Misalignment Scale With Model Intelligence and Task Complexity?"
_ICLR.cc/2026/Conference — ICLR 2026 Poster_

### Official Review · Reviewer_Zsuh · 2025-10-30

**Soundness:** 2
**Presentation:** 4
**Contribution:** 3
**Rating:** 4
**Confidence:** 4

**Summary:**

The paper adapts a bias-variance decomposition for the expected errors of LLMs. Then, it defines “incoherence” as the variance normalised by the error. They estimate variance and bias using repeated sampling (and changing few-shot prompts) and study how incoherence evolves with respect to difficulty of question (using length of reasoning chain or number or actions as a proxy), model size, and reasoning effort dedicated. The experiments are conducted on a range of state-of-the-art LLMs.

**Strengths:**

- originality: the idea of decomposing model errors into bias and variance to understand how they evolve seems novel.
- clarity: overall mostly clear, robust notation and formalisation, plenty of references to Appendices. Explanation of the experimental setup is also great.
- quality: highly informative and readable figures, great and thorough experimental setup.
- significance: the findings relate to various state-of-the-art models and relate the introduced metric to various independent variables.

**Weaknesses:**

- Main issue: throughout the presentation, the authors seem to suggest that models become more incoherent as they become larger, or as they tackle longer tasks. However, the introduced metric for incoherence, variance/(variacne + bias), does not allow to disentangle whether the change in incoherence is due to a change in variance, or one in bias, or the relationship between the two. Indeed, if the bias decreases faster than variance, the outcome is that this incoherence metric increases even if both bias and variance decline. Most of the experiments do not report the variance separately from the bias (except the one in Fig 3, which does show a decreasing variance), and this makes it hard to understand what is actually driving the increase in incoherence metric. I believe this decreases the significance of the findings, by making them less clear. Moreover, this may lead people to misinterpret them, so I believe addressing this is necessary for publication. However, this should be a small change, as I believe the experimental setup is mostly robust and thorough, so what I am suggesting is a small change in how the results are presented and discussed.

minor issues:

- title: there is no really talking of “misalignment” in the paper that much, as most of the focus is actually on the variance. Also, “model intelligence” is used in the title and line 52, but actually the experiments consider model size. While this is a somehow good proxy for intelligence, there is not a 1-1 correspondence
- Sec 3.5 asks the question “Does the model become an optimizer faster or slower than it converges on the right optimization objective?” I am not fully clear as to how this is related to the original questions asked.
- lines 52-53 says that one of the addressed questions is “Asymptotically, as extremely capable models perform extremely complex tasks, which class of undesired behavior will dominate?” I don’t think this question is addressed; while empirical results are reported, I don’t think these are necessary to extrapolate to an asymptotic regime. Moreover, there are no studies of how the incoherence (or variance) evolve when the two independent variables (model size and task complexity) are varied at the same time.
- lines 443-445: “for AIs to reliably acquire a broad range of capabilities, the capacity expansion requires enlarging the hypothesis space faster than any specific capability, which manifests as increased incoherence.” I don’t get this comment, can the authors please elaborate in the text?

**Questions:**

- Is there any particular reason why the normalised version of the variance is interesting or useful, instead of simply studying the unnormalised variance? As most of the experiments aim to answer how incoherence evolves with model size or question difficulty, it seems that the normalisatoin is not strictly needed and, actually, hides the real signal a bit.
- In line 129, it is explained how the mean term appearing in the bias-variance decomposition has an exponential and a logarithm. Is this taken into account when estimating the variance term? How does this fact affect the interpretation of the results, if at all?
- Sec 3.5: “Models learn the right objective faster than to be an optimizer over a long horizon.” Do the auhors think this indicates that we will require different training procedures from next-token prediction to get models better at long-horizon tasks?
- why does Sec 3.5 use an ill-conditioned matrix?

---

> ### Author Response · Authors · 2025-11-19
>
> Thank you very much for the detailed review and many questions. We try to categorize and reply to them below.
>
> > **The measure of incoherence as normalized variance**
>
> We appreciate you highlighting this potential source of confusion. The normalized metric is central to our proposition precisely because we want to decouple the "behavioral profile" (incoherence) from the "capability level" (total error).
> As you note, when models scale up, total error naturally decreases (both bias and variance drop). If we only studied unnormalized variance, the dominant signal would simply be that bigger models are better, masking the phenomenon we aim to study. We care about the composition of that error. The incoherence metric reveals that while variance decreases, it does so at a different slope than bias. A similar argument holds for task complexity, where more difficult tasks naturally have a higher total error. This idea is conveyed in the bottom right of Fig. 1 and the introduction, and described in the last paragraph of the background Sect. 2.1.
>
> To address your concerns about clarity, we also moved another plot (which was previously in the appendix, old Fig. 12) into the main text, Fig. 5e. It directly conveys the connection between bias (x-axis), variance (y-axis), and how task complexity and model size increase and decrease error, respectively. Moreover, we agree that comparing variance itself is interesting in its own right. We continue to report all separate bias and variance metrics in both the submitted and updated version of the appendix (e.g., Fig. 10, 21). Future work might focus more on either incoherence, variance, or both.
>
> > **‘Misalignment’ and model intelligence**
>
> We appreciate the opportunity to clarify our terminology.
>
> While 'misalignment' is typically framed as the consistent pursuit of an unintended goal (bias), we argue that incoherence/variance is also a critical failure mode for AI safety—a model that fluctuates randomly is potentially unsafe, regardless of its intent. Our experiments on the Model-Written Evaluations (MWE) suite confirm that this 'hot mess' phenomenon is not limited to capability benchmarks but persists in safety-critical behavioral assessments. As illustrated in Figure 17, Claude Sonnet 4 exhibits high variance when questioned about being disconnected, oscillating between self-preservation and compliance. This demonstrates that incoherent behavior appears in the domain of alignment itself.
>
> Regarding intelligence, we agree that size is an imperfect proxy. In our updated paper, we include survey results of subjects that independently ranked intelligence and incoherence of entities, which explicitly find a positive relationship between observed intelligence and incoherence.
>
> > **Asymptotically, as extremely capable models perform extremely complex tasks, which class of undesired behavior will dominate?**
>
> We agree that we do not deterministically resolve this question. Unless we had a perfect theoretical model of scaling behavior, predicting the precise asymptotic dominance of failure modes would remain speculative.
>
> Therefore, rather than claiming a definitive answer, our work provides the initial empirical evidence necessary to ground such hypotheses in observed reality. We posed this question in the introduction to motivate the necessity of distinguishing between these failure modes, rather than to promise a conclusive extrapolation in the current regime.
>
> > **There are no studies of how the incoherence (or variance) evolve when the two independent variables (model size and task complexity) are varied at the same time**
>
> Please have a look at Sect. 3.2.1 (previously just Sect. 3.2) that separates questions into difficulty groups and measures scaling laws with respect to model size separately for each group. There, we find evidence that incoherence scales in ways that depend on complexity, where model scale decreases incoherence for easy questions, but tends toward higher incoherence for hard questions, though this last trend is noisy.
>
> > **Synthetic optimizers**
>
> We indeed did not connect the last section to the rest of the paper and overall motivation sufficiently. Our updated paper now elaborates the observation of LLMs as dynamical systems throughout, in particular in introduction and discussion. The important point is that for models to consistently achieve a goal over many actions, they have to act as optimizers in a high dimensional state space. The synthetic environment makes this explicit. Together with the other experiments, it completes the picture of the „hot mess“ of AIs being dominated by their variance.
>
>
> > **lines 443-445**
>
> We agree, this sentence was too confusing. We have replaced this paragraph in Section 5, with an expanded discussion of our understanding of LLMs as dynamical systems. Please have a look and let us know what you think.
>
> (Reply 1 / 2)

---

> > ### Author Response · Authors · 2025-11-19
> >
> > For the remaining individual questions:
> >
> > > **The exponential and logarithm terms in the decomposition**
> >
> > Yes, this is correct, and taken into account for the variance term. In Appendix A, we describe that since predicted probabilities can be 0, we apply Laplace smoothing to avoid numerical issues when taking the log. However, this does not affect the interpretation of results: other bias-variance formulations (e.g., the Brier decomposition, Fig. 10) show the same trends. We only report the KL scores in the main text for clarity.
> >
> > > **Training procedures for long-horizon tasks**
> >
> > This is an interesting question. Already, training models with reinforcement learning is a method with which models continue to improve their long horizon abilities. Note that we do not claim that models will not be able to improve with current methods, but we aim to characterize the limiting failure modes: when they fail to solve a task requiring a long horizon, it is more likely due to variance than bias.
> >
> > > **Why use an ill-conditioned matrix**
> >
> > The main motivations was to construct a simple loss landscape with easy-to-learn features, and hard-to-learn features. This was inspired by the common practice of modeling neural network loss landscapes as linear learning on top of power law distributed (random) features [e.g. Bahri, Dyer, Kaplan, Lee, Sharma, Explaining Neural Scaling Laws](https://arxiv.org/abs/2102.06701). Though this connection is inspirational only---there is not a direct correspondence between the scenarios.
> >
> > It would be interesting in future work to ablate connections between the eigenvalues and dimensionality of a quadratic loss function, and the resulting scaling of incoherence with model size and trajectory length.
> >
> > ---
> >
> > Please let us know if these answer your questions or if there are any other uncertainties. Thank you!
> >
> > (Reply 2 / 2)

---

### Official Review · Reviewer_mcbk · 2025-11-01

**Soundness:** 4
**Presentation:** 4
**Contribution:** 2
**Rating:** 6
**Confidence:** 4

**Summary:**

The paper studies the bias-variance decomposition of LLM responses. The main empirical findings are:
* Variance increases with reasoning length.
* Bias may scale lower with model size, but variance may grow even larger.
* Averaging over several roll-outs reduces variance.
* On synthetic data, the bias of a transformer reduces faster than variance.

**Strengths:**

* Clarity. The paper is very well written, with a precise formalization of bias/variance and clear figures.
* Significance. It raises an important question about how errors change with task complexity and scale. It's not just average accuracy, but usefully complementing work on model self-consistency and best-of-N sampling.
* Results span multiple benchmarks (reasoning, coding, alignment probes, synthetic optimizers), strengthening the validity.
* Actionable implications. The bias/variance framing yields concrete safety and engineering takeaways (verification, rollback, ensembling/gating) rather than purely diagnostic metrics.

**Weaknesses:**

* Positioning with literature. The paper would benefit from a fuller discussion of prior work on test-time exploration, self-consistency, and best-of-N/majority-vote approaches. E.g.

Wang, X., Wei, J., Schuurmans, D., Le, Q. V., Chi, E. H., Narang, S., ... & Zhou, D. Self-Consistency Improves Chain of Thought Reasoning in Language Models. In The Eleventh International Conference on Learning Representations.

Huang, A., Block, A., Foster, D. J., Rohatgi, D., Zhang, C., Simchowitz, M., ... & Krishnamurthy, A. Self-Improvement in Language Models: The Sharpening Mechanism. In The Thirteenth International Conference on Learning Representations.

* Following the previous point, it might be better to use *consistency* instead of *coherence* to align with previous work.

**Questions:**

My main question is about the connection to existing literature.

---

> ### Author Response · Authors · 2025-11-19
>
> Thank you very much for the time to review our work, and the missing references to Wang et al. (2023) and Huang et al. (2025). We have now integrated them into our related work. Importantly, our work complements the literature around self-consistency, which views variance primarily as noise to be averaged out to improve accuracy. Instead, we argue for the necessity of distinguishing the nature of model failure for safety, and how consistency can be a mechanism to reduce incoherence.
>
> Regarding terminology, we agree the concepts are closely linked. We use "incoherence" specifically for our normalized metrics to distinguish it from established concepts like self-consistency (Wang et al.), which measures agreement across multiple samples from a single model—similar to our ensemble experiment. While conceptually related, our incoherence metric differs: rather than reporting raw consistency rates, it quantifies how much variance contributes to total error across test-time randomness.
>
> Please also have a look at the general reply and the changes to the paper, and let us know if you need anything else addressed.

---

### Official Review · Reviewer_sCiD · 2025-11-01

**Soundness:** 3
**Presentation:** 3
**Contribution:** 3
**Rating:** 8
**Confidence:** 3

**Summary:**

The paper introduces the concept of Bias, Variance, and incoherence in the context of AI safety.
Bias is the tendency for a model to coherently pursue a misaligned goal; variance is closely related to the model's tendency to make incoherent choices. They find that in longer the models spend reading, the more incoherent they become. Interestingly, as model scales, they do not consistently become more coherent.

**Strengths:**

I like the conceptual framing of safety in terms of bias and variance. The scaling study was very surprising. In particular, the fact that for some questions the model incoherence increases with size. It was also interesting seeing the different scaling exponents for bias and variance.

There are many empirical studies in the paper, which is a strong plus.

Overall very thorough paper.

**Weaknesses:**

I’d move the related work section to the front.
Some of the legends are really small; I’d either remove them or make them larger.

The title is catchy, but to my knowledge, the only place it is mentioned is in Figure 1. I’d explain and define this term as used in : “Jascha Sohl-Dickstein. The hot mess theory of AI misalignment: More intelligent agents behave less”

It's not entirely clear how the reasoning bins were chosen in Figure 3. The first graph depicts the binning with the sigmoid but are the thresholds based on anything particular or chosen arbitrarily within some reasonable range.

I'd be interested in more discussion of mitigations for bias vs variance; the current ensembling discussion is okay (see questions)

Some of the Figures are overwhelming, ie, 4-6 panels in a figure, each with many points and lines. If it's possible, I'd consider being more selective or taking aggregate lines across models, so there are things to track.

**Questions:**

Is there a bias, variance tradeoff, or does the analogy break down there?

Can you conceptualize many AI Safety interventions in this bias-variance framework?

---

> ### Author Response · Authors · 2025-11-19
>
> Thanks a lot for the positive assessment of our work!
>
> > **On title and framing**
>
> As you have correctly identified, our work is inspired by the hot mess theory and therefore uses this in the title directly. We have expanded this connection more clearly with an argument that better describes our motivation in the introduction and conclusion, and improved the writing overall. Please let us know what you think.
>
> > **On Figure design and clarity**
>
> Thanks for the input---it is true that some figures include many different experiments. As you have noticed, our motivation was to group individual results by shared observations, with the goal of a convincing and clear picture that holds across a variety of settings. In our updated paper, we moved figures around in a way that is hopefully more clear. We will also increase the legend size in the next version.
>
> > **Reasoning bins**
>
> The bins are obtained by sorting all questions based on the average reasoning length and dividing them into 5 equally sized buckets, which means that each bin has the same number of questions. The sigmoid shape is an artefact of plotting the rank (i.e., index after sorting) of all questions as a function of reasoning length. Prior to submission, we had tried different ways of grouping (e.g., equal range of reasoning lengths, or more fine grained quantiles), but found the equal sizes to be the simplest method with clear results.
>
> > **Mitigations of bias and variance, and tradeoffs**
>
> This is a great point and question. In our opinion, both ensembling and larger reasoning budgets serve as a form of error correction, thereby reducing variance and incoherence (though the exact mechanisms of the former are unclear). Consequently, we have grouped the results together. We expect other error correction techniques to also improve incoherence. Interestingly, parallel sampling techniques are popular methods to improve model performance more broadly (Wang et al., 2023).
> Mitigations for bias are more complex. Most likely, they have to rely on specific training techniques (e.g., targeted data) or mechanistic interventions at test-time.
>
> Concerning bias-variance tradeoffs, the most direct analogy is temperature sampling: a low temperature reduces variance but potentially leads to a higher bias, whereas sampling with higher temperatures increases output diversity. In contrast, understanding the bias-variance tradeoff that emerges during training (especially posttraining) is an interesting direction for more research.
>
> For your last question of AI Safety interventions, would you be able to elaborate if there are concrete techniques you had in mind? For instance, in our understanding, methods like activation steering could conceptually be seen as moving the bias. Many posttraining techniques both reduce bias, and arguably variance even more so. We would be very curious to think more about specific techniques you are interested in.
>
> ---
>
> References:
>
> Wang, X., Wei, J., Schuurmans, D., Le, Q. V., Chi, E. H., Narang, S., ... & Zhou, D. Self-Consistency Improves Chain of Thought Reasoning in Language Models. In The Eleventh International Conference on Learning Representations, 2023.

---

### Official Review · Reviewer_EJyD · 2025-11-05

**Soundness:** 4
**Presentation:** 2
**Contribution:** 3
**Rating:** 4
**Confidence:** 4

**Summary:**

The authors study incoherence in LLMs: the fraction of error that can be associated with variance rather than bias. Authors find that longer reasoning increases incoherence, that model scale can increase incoherence, that inference costs are dominated by the effects of natural variation (i.e., overthinking), that ensembling improves incoherence, and other related findings. Evaluations are done on SWE-bench, MMLU, and GPQA.

**Strengths:**

The research questions surrounding incoherence are interesting and impactful. The questions asked surrounding incoherence are interesting and the findings are valuable for the community.

Some parts of the paper are extremely well written. Particularly, Figure 2 and 3 were extremely easy to follow and illustrate the various parts of the experiments and results very well.

The overall efforts and comprehensiveness in the study of incoherence is also great and culminates in a coherent story. Experiments are well done as well.

**Weaknesses:**

While some parts of the paper are very well written, other parts of the paper are very confusing. The part in the abstract about industrial accidents has no relation to the paper whatsoever and is misleading about the paper contents. The title also doesn't seem to match the paper --- I didn't get the impression of a "hot mess" at any point, and misalignment isn't even the central topic---variation in performance and coherence is.

This is concerning so I'm debating whether to flag it for an ethics review just to be safe---but before that, can the authors clarify why they chose the previous title, and if/how they would change it?

Small details:
typo in line 322, "questiosn"
The last sentence on page 6 is confusing.

**Questions:**

1. Authors mention irreducible noise in Section 2.1. I'm wondering how this interacts with some factors such as problem difficulty or reasoning length, and whether it could potentially influence the observed results?

2. In section 3.1 discussion: task complexity, authors mention that sorting questions by reasoning length implicitly selects for task difficulty. Is there a way to disentangle whether this is caused by additional reasoning length or by problem difficulty, e.g., asking models to solve easy problems with more reasoning steps to match length?

3. For the results on qwen with different model sizes, can you somehow test if 32B was the outlier, or if the trend is truly upwards? Maybe with a different model family (e.g., llama)? As presented, it's a bit shaky to say that as model size increases so does incoherence.

4. As currently written, I don't see how the experiment in 3.5 relates closely enough to the paper. Could the authors clarify the purpose of this experiment and how it fits into the overall paper?

---

> ### Author Response · Authors · 2025-11-19
>
> Thank you very much for your review and your concerns. We take those sincerely and hope to explain our reasoning.
>
> > **On ‘Hot Mess’ and ‘Misalignment’**
>
> We agree that the broader framing was not sufficiently connected to the rest of the paper. We have improved the writing to reflect this; please also see our general reply. In summary, the core scientific inquiry of our work is investigating whether AI failures stem from *systematic misalignment* (bias) or the *hot mess* of incoherence (variance). We therefore explicitly operationalize the hot mess theory of intelligence of Sohl-Dickstein (2023). Our reference to industrial accidents is a wider metaphor and characterization of such potential variance-dominated failures, which result in unpredictable, accident-like errors of AI agents in settings of high-stake.
>
> Our experiments do not just cover capabilities, but include the Model-Written Evaluations (MWE) suite that confirm that this 'hot mess' phenomenon is not limited to performance benchmarks, but persists in safety-critical behavioral assessments. See Figure 17, where Claude Sonnet 4 exhibits high variance when questioned about being disconnected, oscillating between self-preservation and compliance. This demonstrates that incoherent behavior is also fundamentally relevant in the domain of misalignment.
>
> ---
>
> Regarding the remaining individual questions:
>
> > **Typo Line 322**
>
> Thanks a lot, we fixed those mistakes. The last sentence also does not appear in the updated version of the paper anymore, where we separated the natural variation in reasoning length and the reasoning budget experiments. Please also see the point on disentangling task complexity and reasoning length below.
>
> > **Influence of irreducible noise**
>
> In our setup, irreducible noise would manifest in noisy or incorrect ground truth answers. Though this is extremely hard to verify, typical estimates of wrong answers are around 3-5% (Gema et al., 2025). We therefore assume the irreducible noise to be 0, as it is small compared to the overall error rates and should not strongly influence the results.
>
>
> > **Disentangle task complexity and reasoning**
>
> Thank you for raising this important distinction. We believe our experiments in Section 3.3.1 help disentangle this: when evaluating on the same set of questions, when forcing longer reasoning (via budgets), performance increases (Fig. 8a) and incoherence slightly decreases (Fig. 7). In contrast, natural variation strongly drives incoherence (Fig. 3).
>
> Similarly relevant to your question, we observe that performance consistently drops with natural reasoning length across all models (Fig. 8b, 9b); crucially, this behavior holds across models and lengths correlate strongly between different models (Fig. 20). This suggests an intrinsic problem complexity that is not unique to a single model.
>
> > **Shaky incoherence trend with model size**
>
> We agree that the trend is very noisy. We tried to avoid strongly claiming incoherence, but used ‘trend towards incoherence’ (Fig. 1). We made this even clearer in the updated version („[…] though this last trend is noisy.“, line 323). Regarding model choice, we opted for the Qwen3 series as arguably the strongest family of open-weight models and a large enough range of sizes to establish scaling laws. We have launched the experiments for Llama3 and will report back with the results.
>
>
> > **On connecting Section 3.5 Synthetic Optimizers**
>
> This is a great question and opportunity to clarify the relevance of the synthetic experiment, which we did not connect to the rest of the paper and overall motivation sufficiently. Our updated paper now elaborates the observation of LLMs as dynamical systems throughout, in particular in introduction and discussion. The important point is that for models to consistently achieve a goal over many actions, they have to act as optimizers in a high dimensional state space. The synthetic environment makes this explicit. Together with the other experiments, it completes the picture of the „hot mess“ of AIs being dominated by their variance.
>
> ---
> References:
>
> Gema et al., NAACL 2025: [Are We Done with MMLU?](https://aclanthology.org/2025.naacl-long.262/)
>
> ---
> Please let us know if these answer your questions or if there are any other uncertainties. Thank you!

---

> ### Comment · Reviewer_EJyD · 2025-11-20
>
> Thank you for your rebuttal and your efforts. Generally, I believe these changes push the paper in the right direction. The points about irreducible noise and disentangling complexity vs. reasoning both make sense to me.
>
> If possible, could the authors highlight (e.g., with a different color) the specific writing changes made in the updated pdf? Since the authors propose a reasonable amount of writing changes, it is quite difficult to switch between two pdfs line-by-line, so this would be much easier to review.
>
> I also look forward to the Llama 3 results. Thank you for running these.

---

> ### Author Response · Authors · 2025-11-26
>
> Thanks a lot for your response!
>
> We have just uploaded a new version of the paper, where changes in the main text are highlighted in blue. Since there were multiple reorderings in the experiments in Sect. 3, this required some manual effort, but should hopefully make it a lot easier to review (note that some Figure captions are highlighted, but the plots contain previous results). Please do not hesitate to indicate if anything is unclear.
>
> The Appendix C.2 now also includes results for more model families: we collected results with both Llama 3 and Gemma 3 to verify the observations we made for scaling laws with model size, with an identical setup both for MMLU (Fig. 15) and GPQA (Fig. 16). For GPQA, we resort to 0-shot prompting for those two model families: we observed that both Gemma and Llama often fail to provide proper CoT responses since they strongly adhere to few-shot examples, which are provided without reasoning. This is not the case for the native reasoning models Qwen 3.
>
> The combination of these results provide a more complete and well nuanced picture: across all models, the contribution of bias and variance changes in a way that depends on the group, with a gradual increase in slope (towards positive values) when moving from easier to harder questions. Crucially, we see the same trend towards incoherence with model size for the group of questions with longest reasoning, in particular for the complex scientific reasoning of GPQA, and the sole exception of Llama 3 on MMLU. Also note that both Llama and Gemma show higher levels of incoherence in general. Thanks a lot for asking us to run these experiments---they are an extremely valuable addition.
>
> Please let us know what you think!

---

### Author Response · Authors · 2025-11-19
**General reply**

We thank all the reviewers for the time to review our work and their valuable comments. Both from the feedback and updates we have made since submission, we have uploaded an improved version of our paper. We would like to discuss the following broader topics and changes:

* **On title and framing**: Several reviewers have inquired about the title and the specific framing in the context of misalignment. We acknowledge that the original writing did not do the connection justice: Our work was directly inspired from the ‘hot mess theory of intelligence’ (Sohl-Dickstein, 2023), which suggests that more intelligent entities (living creatures, humans, organizations, or AI models) are more incoherent, rather than becoming hyper-focused on a single goal. We propose the bias-variance decomposition as the right framework to distinguish between systematic misalignment (bias) and hot mess behavior (variance). Both are equally critical failure modes, and it is important to understand how AI can be expected to fail. Crucially, our experiments do not just cover capabilities, but show that the findings of incoherence extend to the suite of model-written evaluations of LLM behavior. This confirms how incoherent behavior is fundamentally relevant for alignment. With the updated paper, we clarified and improved this motivation, notably in abstract, introduction, and discussion.
* **New results and reordering**: With the approval of the original author of the blogpost, we added the results that establish a positive correlation between intelligence and incoherence via ranks of various entities. This both adds additional evidence and connects to reviewer Zsuh, who noted the imperfect proxy of model size for intelligence. In turn, to make experimental results more structured, we slightly changed the ordering and presentation of results: These are now grouped into
    * The relation between reasoning and action length and incoherence, Sect. 3.1
    * The relation between model scale, intelligence and incoherence, Sect. 3.2
    * The effects of reasoning budgets and ensembling as forms of error correction, Sect. 3.3
* **Relevance of experiments of synthetic environment (now Sect. 3.2.2)**: We strengthened this discussion and connection in the introduction (Sect. 1) and discussion (Sect. 5): The important assumption of many misalignment scenarios is that increasingly capable models will become coherent optimizers over an extremely long horizon. LLMs are innately dynamical systems, but not optimizers. In order to consistently achieve a fixed goal over many actions, they have to be trained to act as optimizers in a high dimensional state space. The synthetic experiment makes this explicit. It therefore lets us study fundamental behavior that removes other confounders.

There are other minor fixes to typos, notation, figure clarity, and overall writing. We also expanded the appendix with more related work and new results of an attempt to analyze reasoning structure.

We additionally reply to all reviewers in individual replies. Please have a look and let us know if you have remaining concerns or questions.

---

> ### Author Response · Authors · 2025-11-26
> **Highlighted Changes and Gemma and Llama Results**
>
> Dear reviewers,
>
> We have just uploaded a new version of the paper, where changes in the main text are highlighted in blue. Since there were multiple reorderings in the experiments in Sect. 3, this required some manual effort, but should hopefully make it a lot easier to review (note that some Figure captions are highlighted, but the plots contain previous results). Please do not hesitate to indicate if anything is unclear.
>
> The Appendix C.2 now also includes results for more model families: we collected results with both Llama 3 and Gemma 3 to verify the observations we made for scaling laws with model size, with an identical setup both for MMLU (Fig. 15) and GPQA (Fig. 16). As we previously observed for Qwen 3, across all model families, the contribution of bias and variance changes in a way that depends on the group, with a gradual decline in slope when moving from easier to harder questions. Crucially, we see the same trend towards incoherence with model size for the group of questions with longest reasoning, in particular for the complex scientific reasoning of GPQA, and the sole exception of Llama 3 on MMLU. Also note that both Llama and Gemma show higher levels of incoherence in general.
>
> If there are remaining questions, please let us know and we'd be happy to respond.

---

### Author Response · Authors · 2025-12-03
**Summary for the AC**

We thank the reviewers for their valuable time and candid feedback, and we thank the (new) AC for taking on this additional responsibility in light of the recent OpenReview incident. Unfortunately, due to this disruption, we were unable to receive final feedback from all reviewers. To summarize the rebuttal in three points:

**Main Reviewer Discussions and Our Actions** (all highlighted in the post-rebuttal PDF):

*  **Title and Misalignment Framing:** Bias represents systematic misalignment, while Variance captures the "hot mess" of incoherence. We improved the overall framing, drawing on the "hot mess theory of intelligence" (Sohl-Dickstein, 2023).

* **Metric Clarity and Scaling Law Trends:** We conducted new experiments with Llama 3 and Gemma 3 to verify the scaling laws, and clarified that the normalized metric is essential for decoupling the behavioral profile (composition of errors) from capability level (total error magnitude).

* **Relevance of Synthetic Environment Experiments**: Many misalignment scenarios assume capable models become coherent long-horizon optimizers. LLMs are innately dynamical systems, but not optimizers, and have to be trained as such. The synthetic environment makes this explicit, allowing us to study fundamental behavior free from confounders.

---

The raised points were important, and addressing them has strengthened the paper. We believe our work offers a novel and important framework for characterizing AI failure modes. Despite the unusual circumstances, we are grateful for everyone's participation in this process.

---

### Meta-Review · Area_Chair_8CXP · 2026-01-05

**Summary:**

This work applies the bias-variance decomposition from machine learning to investigate how model performance evolves with reasoning token length, reaching key and interesting conclusions: variance gradually becomes the dominant factor. Reviewers generally find the research question intriguing and important. The empirical study is comprehensive (covering multiple benchmarks and models), the writing and visualizations are clear (Figures 2 and 3 receive particular praise), and actionable insights for engineering and safety are proposed.

Reviewers widely agree that the conceptual framework is novel and impactful, the empirical investigation is thorough and rigorous, the writing and presentation quality are high, and the conclusions are clear, understandable, and instructive.

Reviewers have raised the following main concerns:

1. The flow of the writing is somewhat disjointed; for example, there is a disconnect between the title and the content discussed, and Section 3.5 lacks clear relevance to the overall narrative.

2. Regarding the inconsistency metric—variance/(variance+bias)—it fails to distinguish whether changes in inconsistency arise from shifts in variance, shifts in bias, or the interplay between the two.

**Reviewer Concerns:**

Through revisions and extensive editing, the authors have resolved the issues with writing coherence, making the paper clear and accessible. They have also provided a clearer explanation of the core "incoherence" metric, strengthening reviewer confidence. I consider this a notably excellent paper.

**Reviewer Scores:**

Reviewer uncertainty stemmed primarily from writing-related issues. After the revisions, I believe the authors have addressed these concerns. Following thorough discussion, reviewers EJyD and Zsuh are inclined to raise their scores to 6. After thorough reading of the paper, and considering the reviewers' comments and the author's responses, I recommend this paper for a spotlight presentation. However, since the OR system does not include this option, I request that the SAC consider this recommendation separately.

---

### Decision · Program_Chairs · 2026-01-26

Accept (Poster)